# A nonstationary analysis for investigating the multiscale variability of extreme surges: case of the English Channel coasts

Imen Turki[1], Lisa Baulon[1,2], Nicolas Massei[1], Benoit Laignel[1], Stéphane Costa[3], Matthieu Fournier[1], Olivier Maquaire[3]

[1] UMR CNRS 6143 Continental and Coastal Morphodynamics`M2C' University of Rouen, 76821 Mont-Saint-Aignan Cedex, France.

[2] French Geological Survey, 3 avenue Claude Guillemin, 45060 Orléans Cedex, France

[3] UMR CNRS 6554  GEOPEN

Corresponding author: Imen Turki (imen.turki@univ-rouen.fr)

## Abstract

This research examines the nonstationary dynamics of extreme surges along the English Channel coasts and seeks to make their connection to the climate patterns at different time-scales by the use of a detailed spectral analysis in order to gain insights on the physical mechanisms relating the global atmospheric circulation to the local-scale variability of the monthly extreme surges. This variability highlights different oscillatory components from the interannual (~1.5-years, ~2-4-years, ~5-8-years) to the interdecadal ( ~12-16-years) scales with mean explained variances of ~ 25 - 32 % and ~ 2 - 4 % of the total variability, respectively. Using the two hypotheses that the physical mechanisms of the atmospheric circulation change according to the timescales and their connection with the local variability improves the prediction of the extremes, we have demonstrated statistically significant relationships of ~1.5-years, ~2-4-years, and ~5-8-years and 12-16-years with the different climate oscillations of

Sea-Level Pressure, Zonal Wind, North Atlantic Oscillation and Atlantic Multidecadal
Oscillation, respectively.
Such physical links have been used to implement the parameters of the time-dependent GEV
distribution models. The introduced climate information in the GEV parameters has
considerably improved the prediction of the different time-scales of surges with an explained
variance higher than 60%. This improvement exhibits their nonlinear relationship with the
large-scale atmospheric circulation.
**Key-Words***:* Coastal extreme surges, multi-timescale variability, climate oscillations,
nonstationary GEV models

# 1. Introduction

Risks assessments has been recognized as an urgent task essential to take effective reduction
of disasters and adaptation actions of climate change. The increase in coastal flood risk is
generally driven by the extreme surges being the result of episodic water fluctuations due to
waves and storm surges. High surges are considered as significant hazards for many low-lying
coastal communities (e.g. Hanson et al., 2011; Nicholls et al., 2011) and are expected to be
intensified with rising global mean sea level (Menendez and Woodworth, 2010).
Being an alarming problem for the coastal vulnerability, extreme events have gained the
attention of the scientists who have reported the dynamics (e.g. Haigh et al., 2010; Idier et al.,
2012; Masina and Lamberti, 2013; Tomasin and Pirazzoli, 2008; Turki et al., 2020) and the
projections (e.g. vousdoukas et al., 2017) of extreme surges considering the stationary and the
nonstationary contributions from tides, waves, sea-level-rise components (e.g. Brown et al.,
2010; Idier et al., 2017), and large-scale climate oscillations (e.g. Colberg et al., 2019; Turki et
al., 2019; 2020).
Under the assumption of a stationary surges, the concepts of return level and return period
provide critical information for infrastructure design, decision-making, and assessing the
impacts of rare weather and climatic events (Rosbjerg and Madsen, 1998).  However, the
frequency of extremes has been changing and is likely to continue changing in the future (e.g.
Milly et al., 2008). Therefore, concepts and models that can account for nonstationary analysis
of climatic and hydrologic extremes are needed (e.g. Cooley, 2013; Salas and Obeysekera,
2013; Parey et al., 2010).
Over the last decade, several studies adopted the nonstationary behaviour of extremes to
estimate their evolution and their return-periods from rigorous models of Extreme Value
Theory (EVT) by incorporating an information related to climate oscillations.
In this way, the recurrence of coastal extreme events over the Northern European continent and
the persistence of high energetic conditions around the Atlantic have been associated with the
deepening of Icelandic Low and the extension/reinforcement of the Azores High. Those facts
can be interpreted, at quasi daily timescale, as the preferred excitation of a given atmospheric
regime close to the positive phase of the North Atlantic Oscillation. The recent predominance
of this regime can be explained partly by the impact of the North Tropical Atlantic Ocean upon
the midlatitude atmosphere and by the increase of greenhouse gas concentration induced by
human activities.
Menendez and Woodworth (2010) have used a nonstationary extreme values analysis together
with the NAO (North Atlantic Oscillation) and Arctic Oscillation (AO) indices for improving
the estimation of monthly extreme sea-levels along the European coasts.
In the Northern Adriatic region, Masina et al. (2013) investigate changes in extreme sea levels
applying a nonstationary approach to the monthly maxima and the climate oscillations of NAO
and AO (Arctic Oscillation) indices. They have suggested that the increase in the extreme water
levels since the 1990s is related to the changes in the wind regime and the intensification of
Bora and Sirocco winds after the second half of the 20th century.
Then, Marcos et al. (2015) have investigated the decadal and multidecadal changes in sea level
extremes using long tide gauge records distributed worldwide. They have demonstrated that
the intensity and the occurrence of the extreme sea levels vary on decadal scales in the most of
the sites in relation with a common large-scale forcing. In the same way, the study of extreme
sea levels along the coastal zones of the North Atlantic Ocean and the Gulf of Mexico has
shown that the mean sea level should be considered as the major driver of extremes (Marcos
and Woodworth 2017) since the intensity of extreme episodes increases at centennial time
scales, together with multidecadal variability.  The extreme sea levels along the United States
coastline between 1929 and 2013 have been investigated by Wahls and Chambers (2015;
2016). Wahls and Chambers (2015) have identified the relation between the multidecadal
variations in extreme sea and the changes in mean sea level. Such relation has been mainly
pointed toward some regions where storm surges are primarily driven by extratropical cyclones
and should contribute in the variation of relevant return water levels required for coastal design.
Such extremes have been then investigated in Wahls and Chambers (2016) works aiming to
define their relationship with the large-scale climate variability by the use of simple and
multiple linear regression models.
In the English Channel, the extreme sea levels have been addressed by several works (e.g.
Haigh et al., 2010; Idier et al., 2012; Tomasin and Pirazzoli, 2008; Turki et al., 2015a; Turki et
al., 2019) with the aim of investigating their dynamics at different timescales and their
connections to the atmospheric circulation patterns.
Haigh et al. (2010) investigated the interannual and the interdecadal extreme surges in the
English Channel and their strong relationship with the NAO index. Their results showed weak
negative correlations throughout the Channel and strong positive correlations at the boundary
along the Southern North Sea. Using a numerical approach, Idier et al. (2012) studied the spatial
evolution of some historical storms in the Atlantic Sea and their dependence on tides.
Recently, Turki et al. (2019) have examined the multiscale variability of the sea-level changes
in the Seine bay (NW France) in relation with the global climate oscillations from the SLP
composites; they have demonstrated dipolar patterns of high-low pressures suggesting positive
and negative anomalies at the interdecadal and the interannual scales respectively.
Despite these important advances, no particular studies exist on sea-level dynamics and
extreme events linked to the large-scale climate oscillations along the English Channel
coastlines. The aforementioned works of Turki et al. (2015a, 2019) have focused on the
multiscale sea-level variability along the French coasts related to the NAO and the Sea-Level
Pressure (SLP) patterns; however, they have not addressed the regional behaviour of the
extreme sea levels in relation with the global climate oscillations.
Then, similar approaches have been used by Turki et al. (2020) to quantify the nonstationary
behaviour of extreme surges and their relationship with the global atmospheric circulation at
different timescales along the English Channel coasts (NW France) between 1964 and 2012.
They have reported that the intermonthly and the interannual variability of monthly extrema
are statistically modelled by nonstationary GEV distribution using the full information related
to the climate teleconnections.
In the same context, the present contribution aims to investigate the interannual and the
interdecadal dynamics of extreme surges along the English Channel coasts (NW France and
SW England) by the use of combining techniques of spectral analyses and probabilistic models.
We hypothesize that different large-scale climate variables may be involved in explaining the
occurrence of extreme surges, and that this dependence can be a function of each timescale.
The rationale behind this hypothesis is based on the following: (1) each timeseries of extreme
surges should depend on different timescales; (2) each timescale should be related to a specific
large-scale oscillation. Using this hypothesis, the linkages between the local extreme surges
and the large-scale climate oscillations are deciphered with the aim to improve the extreme
models using the most consistent large-scale oscillations as covariates.

The overall approach for testing our hypotheses can be described as follows, for a given
extreme surge timeseries: i) identify the short to long timescale oscillations characterizing the
local variability of the extreme surges; ii) explore the correlation between the local extreme
surges and the selected large-scale variable from short to long timescales; iii) select the most
appropriate large-scale variable as an explanatory parameter to be used as a covariate in
nonstationary GEV models and estimate the extreme surges.
The paper is structured as follows. The used hydro-climatic data are presented in section 2,
including local extreme surges and large-scale variables. Section 3 explains the methodological
approach used. Finally, the sections 4 and 5 report the results related to the multiscale
variability of extreme surges along the English Channel and their teleconnections with the
large-scale climate oscillations required for their estimation by the use of GEV extreme models.
The concluding remarks of these findings are addressed in section 6.
## 2. Database description
The present research focuses on the dynamics of extreme surges along the English Channel
coasts (French and the Britannic coasts); It has been conducted in the framework of some
French research programs: RICOCHET (ANR program), RAIV COT (Normandy Region
program) and the international project COTEST (CNES-TOSA program) related to the future
mission Surface Water and Ocean Topography (SWOT).
The English Channel (Figure 1) is a shallow sea between Northern France and South England,
connecting Atlantic Ocean to North Sea. Melting of retreating glaciers formed a megaflood in
the southern North Sea and it geographically separated Britain from Europe and formed English
Channel at the last Quaternary Period (Collier et al. 2015). English Channel has a complex sea
floor due to its characteristics of formation. It is deep and wide on the western side, narrower
and shallower towards Strait of Dover. Largest width of the Channel is around 160 km (Figure
1). The average depth of the channel is about 120 m. It gradually narrows eastward to a width
of 35 km and depth of around 45 m in the Dover Strait. The east to west extent of the Channel
is about 500 km. The overall width of shallow depths is wider in the French side of the channel.
The extreme storm surges of this area are mostly occurred by low pressure systems from the
Atlantic Ocean, propagating eastwards or storm surges propagating south from the North Sea
(Law, 1975). The area is exposed to major storms from Atlantic side of the channel, having a
maximum fetch of winds, from west to southeast then to northwest.
Three tide gauge sites along the French coasts have been used in the present study: (1) Dunkirk
station which is a few kilometres away from Belgian borders, (2) Cherbourg station located on
the Cotentin Peninsula and at the opening of the Atlantic Sea, (3) Brest station which is a
sheltered bay located at the western extremity of metropolitan France and connected to the
Atlantic Ocean.

*Figure 1 Geographical location of the study area and the different tide gauges along the*
*English Channel coasts: Brest, Cherbourg, Dunkirk (NW France); Dover and Weymouth (SW UK).*

Two tide gauge sites along the Britannic coats have been used: (1) Dover station which is

separated from Dunkirk by the North Sean and (2) Weymouth station symmetrically with

respect to Cherbourg.

The French tide gauges are operated and maintained by the National French Center of

Oceanographic Data (SHOM) while the Britannic tide gauges are operated by the British

Oceanographic Data Center. All stations are referenced to the hydrographic zero level; they

provide time-series of hourly observations measurements until 2018.

Available data are summarized as the following: Brest (168 years between 1850 and 2018);

Cherbourg and Dunkirk (54 years between 1964 and 2018); Dover (53 years between 1963 and

2018); Weymouth (28 years between 1990 and 2018).

The hourly measurements suffer from some gaps of daily length distributed along the time-

series. These gaps have been processed by the hybrid model for filling gaps developed by Turki

et al. (2015b) by using the SLP as covariate in ARMA methods and the memory effects of the

previous distribution of surges to estimate the missing values and fill the gaps. This model has

been used in the recent works of Turki et al., (2019; 2020).

The large-scale atmospheric circulations are represented in this work by four different climate
indices which are considered as fundamental drivers in the Atlantic regions (Massei et al.,
2017; Turki el al., 2019; 2020): the Atlantic Multidecadal Oscillation (AMO), the North
Atlantic Oscillation (NAO), the Zonal Wind (ZW) component extracted at 850hPa, and the
Sea-Level Pressure (SLP).
Monthly time-series of climate indices have been provided by the NCEP-NCAR Reanalysis
fields   (http://www.esrl.noaa.gov/psd/data/gridded/data.ncep.reanalysis.derived.html)   until
2017. The different indices have been extracted  during the same period of the sea-level
observations at the four stations Cherbourg, Dunkirk, Dover and Weymouth. For the longest
timeseries of Brest (1850 - 2018), the use of climate indices has been limited according to their
initial date availability (AMO: 1880 – 2017; NAO: 1865-2017; SLP: 1948-2017; ZW: 1865-

2017).

**3. Methodological Approach**
**2. 1 Extraction of residual sea level: 'surges'**

The total sea-level height, resulting from the astronomical and the meteorological processes,
exhibits a temporal non-stationarity which is explained by a combination of the effects of the
long-term trends in the mean sea level, the modulation by the deterministic tidal component
and the stochastic signal of surges, and the interactions between tides and surges. The
occurrence of extreme sea levels is controlled by periods of high astronomically generated
tides, in particular at inter-annual scales when two phenomena of precession cause systematic
variation of high tides. The modulation of the tides contributes to the enhanced risk of coastal
flooding. Therefore, the separation between tidal and non-tidal signals is an important task in
any analysis of sea-level time-series. By the hypothesis of independence between the
astronomical tides and the stochastic residual of surges, the nonlinear relationship between the
tidal modulation and surges is not considered in the present analysis. Using the classical
harmonic analysis, the tidal component has been modelled as the sum of a finite set of sinusoids
at specific frequencies to determine the determinist phase/ amplitude of each sinusoid and
predict the astronomical component of tides. In order to obtain a quantitative assessment of the
non-tidal contribution in storminess changes, technical methods based on MATLAB t-tide
package have been applied to the seal level measurements, demodulated from long-term
components (e.g. mean sea level, vertical local movement ), for estimating year-by-year tidal
constituents. A year-by-year tidal simulation (Shaw and Tsimplis, 2010) has been applied to
the sea-level time-series to determine the amplitude and the phase of tidal modulations using
harmonic analysis fitted to 18.61-, 9.305-, 8.85-, and 4.425-year sinusoidal signals (Pugh,
1987). The radiational components have been also considered for the extraction of the
stochastic component of surges (Williams et al., 2018).
## 3.2 Wavelet spectral analysis
The Continuous Wavelet Transform CWT is generally used for data analysis in hydrology,
geophysics, and environmental sciences (Labat, 2005; Sang, 2013; Torrence and Compo,
1998). This technique produces the timescale with the means of the Fourier transform contour
diagram on which the time is indicated on the x-axis, the timescale (period,) on the y-axis, and
the variance (power) on the z-axis.
Then, a wavelet multiresolution analysis has been used to decompose the signal of monthly
extreme surges into different internal components corresponding to different timescales. This
decomposition consists on applying a series of iterative filtering to the signal by the use of low-
pass and high-pass filters able to produce the spectral components describing the total signal.
More details are presented in the recent works of Massei et al. (2017) and Turki et al., (2019).
In summary, the total signal has been separated into a relatively small number of wavelet
components from high to low frequencies that altogether explains the variability of the signal;
this will be illustrated later using the hourly measurements and the monthly maxima of surges.
The wavelet coherence has been calculated to investigate the relationship between the extreme
surges and the climate oscillations by identifying the timescales where the two timeseries co-
vary, even if they do not display high power. Here, a significance test has been implemented
by the use of a Monte Carlo analysis based on an autocorrelation function of two timeseries (
Grinsted et al., 2004).


*3. 3 Stationary and Nonstationary extreme value model*
Finally, and with the aim of addressing the nonstationary behaviour of extreme surges, the
monthly maxima of the surges have been calculated and decomposed with the multiresolution
analysis. Then, a nonstationary extreme value analysis based on the GEV distribution with
time-dependent parameters (Coles, 2001) has been implemented to model the series of the
monthly maxima surges. There are several GEV families which depend on the shape parameter,
e.g. Weibull ($\varepsilon < 0$), Gumbel ($\varepsilon = 0$), and Fréchet ($\varepsilon > 0$). The three parameters of the GEV (i.e.
location $\mu$, scale $\psi$, shape $\varepsilon$) are estimated by the maximum likelihood function.
The nonstationary effect was considered by incorporating the selected climate indices (NAO,
AMO, ZW, and SLP) into the parametrization of the GEV models. Akaike Information
Criterion (AIC) has been used to select the most appropriate probability function models. The
methods of maximum likelihood were used for the estimation of the distribution's parameters.
The approach used considers the location ($\mu$), the scale ($\psi$), and the shape ($\varepsilon$) parameters with
relevant covariates, which are described by a selected climate index:
$$\mu(t) = \beta_{0,\mu} + \beta_{1,\mu}Y_1 + \cdots + \beta_{n,\mu}Y_n \quad (1)$$

$$\psi(t) = \beta_{0,\psi} + \beta_{1,\psi}Y_1 + \cdots + \beta_{n,\psi}Y_n \quad (2)$$

$$\varepsilon(t) = \beta_{0,\varepsilon} + \beta_{1,\varepsilon}Y_1 + \cdots + \beta_{n,\varepsilon}Y_n \quad (3)$$

Where $\beta 0$, $\beta 1$,…, $\beta n$ are the coefficients, and $Y_i$ is the covariate represented by the climate
index. For each spectral component, only one climate index can be used to be introduced into
the parameters $\mu$, $\psi$, and $\varepsilon$ of the nonstationary GEV model (into one of them, into two of them
or into the three parameters).
With the aim of optimizing the best use of the most appropriate climate index (detailed in
section 3.4) into the different GEV parameters, a series of sensitivity analyses were
implemented for each timescale. The AIC measures the goodness of the fitting of the model
(Akaike, 1973) to the relation AIC =-2l+2K; where l is the log-likelihood value estimated for
the fitted model, and K is the number of the model parameters. Higher ranked models should
result from lower AIC scores.
The non-stationary return levels and return-periods have been calculated using Bayesian
inference, implemented in the Non-stationary Extreme Value Analysis (NEVA) software R-
package.  The Confidence intervals for the return level estimates have been calculated by the
use of the method of delta (Coles, 2001).

### *3. 4 Determination of the most appropriate climate oscillation*
### *connected to each timescale extreme surges for GEV models*

As suggested previously, the main hypothesis presented in this research is that effects of the
physical mechanisms on the extreme surges vary according to the timescale and each scale
should be related to a given climate oscillation.
This hypothesis has been investigated by two approaches:
(1) a spectral approach based on the use of wavelet techniques (wavelet multiresolution and
wavelet coherence as detailed in section 3.2) for optimizing the physical relationship between
the climate index and the extreme surges at each timescale.  Here, a bootstrap approach has
been applied to assess the statistical significance of the correlation between the spectral
component of the extreme surges and the climate oscillation at each timescale. By resampling
the timeseries 10.000 times, 95% confidence intervals have been considered to extract the best
climate information fitting the extreme surges (Villarini et al., 2009).
Here, the confidence intervals (CI) have been calculated by the bootstrap technique by
simulating the monthly maxima of surges (spectral component) from the climate index
(spectral component) at each timescale (new samples with a size of 1000). When the original
surges have been fitted to the simulated ones, 95% confidence intervals for the maximum
likelihood estimates have been calculated.
(2) a Bayesian estimation has been used to make inferences from the Likelihood function. The
reason behind the choice of this approach is overcoming the limitation of short time-series with
small size, the case of Weymouth station where the measurements covers the period from 1991
and 2018. A technique of Markov Chain Monte Carlo (MCMC), implemented in the evbayes
package within R software, has been used basing on multiple simulations (the number of
simulations is varying as a function of the length of the timeseries).
For each spectral component, a sample of 100.000 simulations has been modelled by GEV
using a given climate index. The upper and lower quantiles of the posterior probability
distribution for the parameters of the MCMC sample are taken. The goodness of fit has been
taken as a function of the values of the upper and the lower quantiles; best results have been
considered when these values are higher than 92.5% and lower than 5.2%, respectively.

### 304    4. Multi-timescale variability of extreme surges

The variability of the monthly extreme surges along the English Channel coasts has been
investigated using the continuous wavelet transform (CWT). In the spectrum of Figure 2, the
colour scale represents an increasing power (variance) from red to blue and pink. The CWT
diagrams highlight the existence of several scales for all sites with different ranges of
frequencies: the interannual scales of ~ 1.5-yr, ~ 2-4-yr, ~ 5-8-yr and the interdecadal scale of
~ 12-16-yr.
*Figure 2. CWT of monthly maxima of surges in Brest, Cherbourg, Dunkirk, Dover and*
*Weymouth.*
The variability of surges is clearly dominated by the interannual frequencies (~ 1.5-yr, ~ 2-4-
yr, ~ 5-8-yr) explaining a mean variance between 32% and 25% of the total energy (Table 1).
In Dover and Weymouth, the low frequencies of ~ 2-4-yr are well-structured with a mean
explained variance of 9.5% while it is of 7% for ~ 5-8-yr. These percentages decrease slightly
for the French sites to 8% and 5%, respectively. At ~ 1.5-yr, the explained variance is higher
than 16% and 13% respectively in Britannic and French coasts. The interdecadal frequency of
~ 12-16-yr varies between 2% and 4% from the total signal. This frequency is not observed in
the shortest timeseries of Weymouth (Table 1).
The interannual variability (time-scales higher than ~1 year) seems to be highly represented in
the monthly extrema CWT (Figure 2). It's not the case for the monthly mean surges (Figure
3.a) where most power spectrum is concentrated on the annual cycle with an explained variance
higher than 50%.
*Table 1. The explained variance expressed as percentage of total variance of monthly extreme surges.*

The time-dependent PDF of the monthly mean and maximum surges over a period of 10 years,
for illustration purpose, is displayed in Figure 3.b. The ~1-yr component of monthly mean
surges is largely manifested with a pronounced variation of the Gaussian curves in time; such
variations take wavelengths of approximately ~2-yr and ~4-yr. This result exhibits that the
interannual frequencies of ~2-yr and ~4-yr are modulated within the annual mode for the mean
surges while they are implicitly quantified for the monthly maxima.
*Figure 3. Multiscale variability of the monthly mean and maximum surges in Brest. (a) CWT*
*of monthly mean surges; (b) Interannual variability of monthly and extreme surges*

Results have been explored to investigate the nonstationary dynamics of surges at different
timescales. We have applied the wavelet multiresolution decomposition of monthly extrema
for each site. The process has resulted in the separation of several components with different
time-scales. Only the wavelet components, with have been considered in this work. In this
research, we are interested in the time-scales higher than 1 year, i.e. traduced by three
interannual scales (~ 1.5-yr, ~ 2-4-yr, and ~ 5-8-yr) and a interdecadal scale of ~ 12-16-yr. We
focused only on the interannual and the interdecadal scales whose fluctuations correspond to
the oscillation periods less than half the length of the record and exhibit a high-energy
contribution on the variance of the total signal. The lowest frequency, corresponding to ~ 12-
16-yr is easily calculated from the longest record of Brest.
*Figure 4 Wavelet details (components) resulting from the multiresolution analysis of surges*
*at the interannual (~ 1.5-yr , ~2-4-yr and ~5-8-yr) and interdecadal (~12-16-yr) time scales*
*for all sites (Brest, Cherbourg, Dunkirk, Dover and Weymouth).*

Figure 4 shows a series of oscillatory components of surges from interannual to interdecadal
scales, not easily quantified by a simple visual inspection of the signal. High similarities
between the different sites have been highly observed for the interannual and the interdecadal
scales of ~ 5-8-yr and ~12-16-yr while they are less pronounced at the small scales of ~ 1.5-yr.
At this timescale, the differences in the extreme surges can be explained by local physical
phenomena controlling their dynamics. Such processes are mainly induced by combining the
effects of meteorological and oceanographic forces including changes in atmospheric pressures
and wind velocities in shallow water areas. Beyond ~ 1.5-yr, the variability of extreme surges
at larger scales seems to be quite similar in terms of frequency and amplitude for the five sites.
Such large variability reveals the physical effects of a global contribution related to climate
oscillations. The extent of the large-scale oscillations is not strictly similar and changes
according to the timescale variability since the dynamics of surges is not necessarily related to
the same type of atmospheric circulation process. This relationship will be addressed later in
the second part of this section.
Here, the multiscale variability of extremes has been investigated from the spectral components
of surges along the English Channel coasts. This signal has been linearly extracted from the
total sea level, provided by tide gauges, by the use of the classical harmonic analysis and thanks
to the assumption that the water level is the sum of the mean sea level, tides, and surges. This
assumption approximates the quantification of both components in the English Channel where
the significant tide-surge interactions (Tomassin & Pirazzoli, 2008) and the effects of the sea-
level rise on tides and surges are important (e.g. Idier et al., 2017). Neglecting this nonlinear
interaction between the surges, tides, and the sea-level rise suggests some uncertainties in the
estimation of the high frequencies of the spectral components between daily and monthly
scales, which is not the focus of the present work where the interannual and the interdecadal
scales are investigated.
Similar interannual timescales have been observed along the French coasts of Dunkirk, Le
Havre and Cherbourg in Turki et al., (2020) works where the intermonthly and the interannual
variability of 48-year hourly surges has been investigated. They have demonstrated that the
timescales smaller than ~ 1.5-yr are differently manifested between the different sites. These
differences have been associated to the local variability of surges induced by combining the
effects of meteorological and oceanographic forces including changes in atmospheric pressures
and wind velocities in shallow water areas. As demonstrated in Turki et al. (2020) works, the
mean explained variance of the interannual fluctuations (~ 1.5-yr, ~ 2-4-yr, and ~ 5-8-yr) is
around 25% of the total surges along the French coasts (Table 1). This value is higher than 32%
in Weymouth and Dover while the explained variance of the interdecadal scales (~ 12-16-yr)
is also more important with 3.5% (compared to 2% for the French coasts).

The interdecadal variability (~ 12-16-yr) of extreme surges have been evidenced by Turki et
al., (2019) in the Seine bay (NW France). Strong physical relations have been exhibited
between the interdecadal time of ~ 12-16-yr and the exceptional stormy events produced with
surges higher than 10-year return period level. The connections between the low-frequency
components and the historical record of the exceptional events suggested that storms would
occur differently according to a series of physical processes oscillating at multi-timescales;
these processes control their frequency and their intensity (Turki el al., 2019).
.Accordingly, the multiscale variability of extreme surges exhibits a nonstationary behaviour
modulated by a non-linear interaction between the different interannual and the interdecadal
timescales. Then, assessing the effect of the nonstationary behaviour at different timescales is
important for improving the estimation of extreme values and the projection of storm surges.

**5. Large-scale climate North-Atlantic oscillations and their link to**
**extreme surges in the English Channel**

In this part, a new hybrid approach combining the spectral analysis and the nonstationary GEV
models has been used to investigate the connection between the multi-timescale variability of
local surges and the large-scale climate North Atlantic oscillations.
As proposed by Turki et al. (2019; 2020), the hypothesis used in the present work is that the
multi-timescale variability of the local extreme surges should be strongly related to different
climate teleconnections induced by a complex contribution of many physical mechanisms. This
non-linear relationship varies according to each timescale which depends on a specific large-
scale oscillation of atmospheric circulation.

*5.1 To what extent would large-scale climate oscillations link extreme surges?*
The wavelet coherence (WC) diagrams between the monthly maxima of surges and the
different climate indices of SLP, ZW, NAO, AMO, introduced previously as the main
atmospheric circulation within the English Channel, are illustrated respectively in Figures 5, 6,
7 and 8. Results provided by these diagrams highlight:
1. The connection between the climate oscillations and the extreme surges is manifested

differently as a function of the timescale. From a visual inspection of the different

spectra provided WC, the most significant correlations of extreme surges have been

identified with SLP, ZW, NAO and AMO respectively at ~ 1.5-yr, ~ 2-4-yr, ~ 5-8-yr

and ~ 12-16-yr.

2. Each timescale exhibits mainly strong links with its associated climate index (explained

variance varying between 55% and 80%) and weak ones with other indices (explained

variance varying between 15% and 5%). Table 2 summarizes the contribution of the

different climate oscillations in the different interannual and interdecadal timescales of

extreme surges. Here, mean values between the different sites are presented.

For example, SLP diagrams reveal significant relationships with ~ 1.5-yr surges (well-
structured forms with high concentration of pink-blue colours in Figure 5); limited
correlations, locally positioned in time, have been observed at ~ 2-4-yr and ~ 5-8-yr scales.
ZW shows strong correlations with interannual surges at ~ 2-4-yr (blue to pink colour at
this scale; Figure 6) and others correlations at smaller and larger timescales of ~ 1.5-yr and
~ 5-8-yr, respectively. Similarly, NAO presents high links with ~ 5-8-yr surges and small
relations with ~ 2-4-yr and ~ 1.5-yr (Figure 7).
*Figure 5. Cross-wavelet correlations between monthly extrema of surges and Sea Level*
*Pressure (SLP).*
*Figure 6. Cross-wavelet correlations between monthly extrema of surges and Zonal Wind*
*(ZW).*

*Figure 7. Cross-wavelet correlations between monthly extrema of surges and North Atlantic*
*Oscillation (NAO).*

*Figure 8. Cross-wavelet correlations between monthly extrema of surges and Atlantic*
*Multidecadal Oscillation (AMO).*

The ~ 1.5-yr scale highlights strong correlations with SLP with an explained variance of 75%;
25% of this scale should be explained by the influence of other climate oscillations (basically
ZW and NAO with a mean explained variance of 10% and 6%, respectively; Table 2) and the
combining effects of local driven forcing induced by winds and waves.
*Table 2. The mean explained variance expressed as percentage of total variance provided by the*
*wavelet coherence between the extreme surges and the climate Oscillations (SLP, ZW, NAO, AMO).*

65% of ~ 2-4-yr scale is correlated with ZW while 5% and 12% is explained by the effect of
NAO and SLP, respectively. The effects of NAO on the ~ 5-8-yr vary between 55% and 65%;
minor influence at this scale has been observed with SLP and ZW explaining a mean variance
of 13%. The interannual scales of surges are slightly influenced by AMO oscillations with low
values of variance lower than 1% (Table 2).
At interdecadal scales of ~ 12-16-yr, the extreme surges are mainly controlled by the AMO
oscillations with a mean explained variance of 80% while the effects of NAO is limited to 10%.
Figure 9 displays the spectral components of the four climate oscillations, provided by a multi-
resolution analysis, together with the spectral components extracted from the extreme surges
(Figure 4) with the aim to quantify the different connections between both variables at the
interannual and the interdecadal timescales.
For each timescale, a bootstrap approach has been applied to assess the statistical significance
of the correlation between the spectral component of the extreme surges and the climate
oscillation (table 3). By resampling the timeseries 10.000 times, 95% confidence intervals have
been considered to extract the best climate information fitting the extreme surges (Villarini et
al., 2009).

The best correlation of each surge component (i.e. ~ 1.5-yr, ~ 2-4-yr, ~ 5-8-yr and ~ 12-16-yr)
with the suitable climate index  ( i.e. SLP, ZW, NAO and AMO)  is illustrated in Figure 9.
The interannual and the interdecadal variability of extreme surges and their multiscale
connection with the climate oscillations highlight the nonlinear relationship between large- and
local- scales.
Therefore, the interannual and the interdecadal extreme surges have proven to be strongly
related to different composites of oscillating atmospheric patterns. Such composites seem to be
not necessarily similar for the different timescales. The use of a multiresolution approach to
investigate the dynamics of the extreme surges into the downscaling studies proves to be useful
for assessing the nonstationary dynamics of the local extreme surges and their nonlinear
interactions with the large-scale physical mechanisms related to climate oscillations.
Investigating the complex relationships between the climate oscillations and the multi-
timescale surges has exhibited a multimodel climate ensemble that should be used to better
understand this complexity.
The interannual connections between the local hydrodynamics and the climate variability have
been investigated in numerous previous works focused on the atmospheric circulation with
different related mechanisms (e.g., Feliks et al., 2011; Lopez-Parages et al. 2012; Zampieri et
al., 2017). As demonstrated by the recent works of Turki et al. (2019, 2020), the effects of SLP
oscillations on the ~ 1.5-yr variability of extreme surges are described by dipolar patterns of
high-low pressures with a series of anomalies which are probably induced by some physical
mechanisms linked to the North-Atlantic and ocean/atmospheric circulation oscillating at the
same timescale.
The SLP fields combined with the baroclinic instability of wind stress have been related to the
Gulf Stream path as given by NCEP reanalysis (Frankignoul et al., 2011); the dominant signal
is a northward (southward) displacement of the Gulf Stream when the NAO reaches positive
(negative) extrema. Daily mean SLP fields have been used by Zampieri et al. (2017) to analyse
the influence of the Atlantic sea temperature variability on the day-by-day sequence of large-
scale atmospheric circulation patterns over the Euro-Atlantic region. They have associated the
significant changes in certain weather regime frequencies to the phase shifts of the AMO. For
hydrological applications, several works have investigated the multiscale relationships between
the local hydrological changes and the climate variability. Lavers et al. (2010) associated the
7.2-yr timescales to SLP patterns which are not exactly reminiscent of the NAO and define
centers of action which are shifted to the North.
Regarding the ZW (u850), results have shown its correlation with the interannual scales of ~2-
4-yr extreme surges as suggested also in the recent findings of Turki et al. (2020). Its influence
has been proven also at smaller (~1.5-yr) and larger scales (5-8-yr). Additionally, to extreme
surges, the interaction between the ZW and the temperature at different timescales has been
highlighted in some previous researches (e.g., Andrade et al., 2012; Seager et al. 2003;
Woodworth et al., 2007). Along UK and Northern English Channel coats, Changes in trends
of extreme waters and storm surges have been explained by variations of energy pressure and
ZW variability additional to thermosteric fluctuations linked to NAO (Woodworth et al., 2007).
Andrade et al. (2012) have used the component of ZW at 850 hPa to investigate the positive
and the negative phases of the extreme temperatures in Europe and their occurrence in relation
with the large-scale atmospheric circulation. They suggested that both phases are commonly
connected to strong large-scale changes in zonal and meridional transports of heat and
moisture, resulting in changes in the temperature patterns over western and central Europe
(Corte-Real et al., 1995; Trigo et al., 2002). The physical connections between ZW and the
extreme events from 11 Global Climate Model runs have been demonstrated by the studies
from Mizuta (2012) and Zappa et al (2013); they have suggested the complex relationship
between the climate oscillation and the jet stream activity. They have found a slight increase
in the frequency and strength of the storms over the central Europe and decreases in the number
of the storms over the Norwegian and Mediterranean seas.
The NAO is considered as an influencing climate driver for the large-scale atmospheric
circulation, as suggested by other researches (e.g. Marcos et al., 2009; Philips et al., 2013).
The existence of long-term oscillations originating from large-scale climate variability and thus
controlling the interannual extreme surges has been highlighted from investigating the low
frequencies of the sea levels along the English Channel. This is in agreements with the results
recently demonstrated by Turki et al. (2020) and the present finding exhibiting the strong links
between NAO oscillations and the ~ 5-8-yr extreme surges along the English Channel coasts.
The physical mechanisms related to the effects of the continuous changes in NAO patterns on
the sea-level variability have been addressed in several studies (e.g., Marcos et al., 2005;
Tsimplis et al., 1994). At the interannual scales, the key role of NAO on the sea-level variability
has been explained by some previous works: Philips et al. (2013) investigated the influence of
the NAO on the mean and the maximum extreme sea levels in the Bristol Channel/Severn
Estuary. They have demonstrated that when high NAO winters increase in the positive phase,
wind speeds also escalate while increasing the negative NAO warmers results in low wind
speeds. Then, the correlation between the low/high extreme surges and the NAO in the Atlantic
has demonstrated a proportionality between NAO values and the augmentation in the winter
storms. Feliks et al. (2011) defined significant oscillatory timescales of ~ 2.8-yr, ~ 4.2-yr, and
~ 5.8-yr from both observed NAO index and NAO atmospheric marine boundary layer
simulations forced with SST; they have suggested that the atmospheric oscillatory modes
should be induced by the Gulf Stream oceanic front.
Strong correlations between the monthly extreme surges and the AMO oscillations have been
identified at the timescale of 12-16-yr (Figure 8 and Figure 9; in particular for Brest). Since the
period of 1990's, the AMO and the extreme surges oscillate in opposition of phase. This shift
should be explained by a substantial change in European climate manifested by cold wet and
hot dry summers in the northern and the southern Europe, respectively; as discussed by Sutton
and Dong (2012). They have demonstrated that the patterns, identified from the European
climate change around 1990's, are synchronised with changes related to the North Atlantic
Ocean.
Other weak links with the AMO have been identified at the interannual timescales of ~5-8-yr.
along the studied sites. In agreement of previous works (e.g., Enfield et al., 2001; Zampieri et
al., 2013; 2017), the effects of the AMO oscillations are mainly manifested at the interannual
timescales to control the variability of hydrological (e.g. rainfall) and oceanographic (e.g.
surges) variables. Generally, the climate oscillations of AMO are associated to the SST
variability with a time cyclicity of about 65-70-years (e.g. Delworth and Mann, 2000; Enfield
et al., 2001). During the warming periods of the 1990's, the AMO shifts from the negative to
the positive phases in the Northern Hemisphere corresponding to cold and warm periods (e.g.
Gastineau et al., 2012; Zhang et al., 2013). This shift can be responsible on changes in the
hydrodynamic conditions (e.g. Zampieri et al., 2013).
The influence of the AMO oceanic low frequencies in the modulation of the mechanisms of
the atmospheric teleconnections at the interannual timescales has been investigated in many
previous works (e.g Enfield et al. 2001). At decadal timescales, the existing relationships
between the winter NAO and the AMO variability is more complex (e.g. Peings and
Magnusdottir, 2014).
The effects of the AMO-driven climate variability on the seasonal weather patterns have been
investigated by Zampieri et al. (2017) in Europe and the Mediterranean. They have
demonstrated significant changes in the frequencies of weather regimes involved by the AMO
shifts which are in phase with seasonal surface pressure and temperature anomalies. Such
regimes, produced in Spring and Summer periods, are differently manifested in Europe with
anomalous cold conditions over Western Europe (Cassou et al., 2005; Zampieri et al., 2017).
In summary, four atmospheric oscillations have proven to be significantly linked to the
interannual and interdecadal variability of extreme surges. This physical link varies according
to the timescale exhibiting a nonlinear interaction of the same oscillations with other scales.
Such nonlinear behavior depends on the dynamics of the different sequences of the atmospheric
and water vapour transport patterns during the month prior to the sea-level observations (e.g.
Lavers et al., 2015). As suggested by Turki et al. (2020), the atmospheric circulation acts as a
regulator controlling the multiscale variability of extreme surges with a nonlinear connection
between the large-scale atmospheric circulation and the local scale hydrodynamics. This multi-
timescale dependence between the local extreme dynamics and the internal modes of climate
oscillations is still under debate. Understanding these physical links, even their complexities,
are useful to improve the estimation of the extreme values in coastal environments; which is
the objective of the next part.

## *5.2 Nonstationary modelling of extreme surges*

In this part, stationary and nonstationary extreme value analyses based on GEV distribution
with time-dependent parameters (Coles, 2001) have been implemented to model separately the
different spectral components of extreme surges. Four GEV stationary (GEV0) and
nonstationary (GEV1, GEV2 and GEV3) models have been applied to each timescale and each
site. The GEV distribution uses the maximum likelihood method by parametrizing the location,
scale, and shape of the model. We have used the 'trust region reflective algorithm' for
maximizing the log-likelihood function (Coleman and Li, 1996).
The connections between the climate oscillations and the monthly maxima at the different
timescales (Figure 9)., presented previously (section 5.1), have been explored as a first
hypothesis for the implementation of the nonstationary GEV models. Indeed, multiple
simulations of Markov Chain Monte Carlo (MCMC) techniques based on Bayesian approaches
have been employed for extreme surge components (i.e. ~ 1.5-yr, ~ 2-4-yr, ~ 5-8-yr and ~ 12-
16-yr provided by the multiresolution wavelet decomposition) to identify the best covariates of
climate oscillation for parametrizing the nonstationary GEV models. The most of simulations
has mainly supported the results outlined in the previous section: the ~ 1.5-yr of SLP, ~ 2-4-
yr of ZW, ~ 5-8-yr of NAO and ~ 12-16-yr of AMO oscillations are considered as the best
covariates for modelling respectively the ~ 1.5-yr, ~ 2-4-yr, ~ 5-8-yr and ~ 12-16-yr of monthly
extreme surges.
Once the climate covariate has been selected for each timescale, three nonstationary models
have been used by introducing the climate information as a covariate into: (1) the location
parameter (GEV1); (2) both location and scale parameters (GEV2); (3) all location, scale and
shape parameters (GEV3). The structure of the most appropriate nonstationary GEV
distribution has been selected by choosing the most adequate parametrization that minimizes
the Akaike information criterion (Akaike, 1974). The goodness of fit for each model has been
checked through the visual inspection of the quantile-quantile (Q-Q) plots (Figure 10); these
plots compare the empirical quantiles against the quantiles of the fitted model. Any substantial
departure from the diagonal indicates inadequacy of the GEV model.

At the interannual scales and for all sites, results provided by the nonstationary GEV1-3 reveal
a better performance (the lowest values of AIC) of extreme estimation compared to the
stationary models of GEV0 and give the most appropriate distributions by the use of the climate
large-scale covariates for specific oscillating components of extreme surges. Nevertheless, this
improvement from the stationary to the nonstationary models has not been clearly observed for
the interdecadal scales where the extreme estimation, provided by the different GEV models,
is very similar (Table 4).  The lowest values of AIC have been shown by GEV3 for ~1.5-yr,
GEV2 for ~2-4-yr and GEV1 for ~5-8-yr (Table 4). The Q-Q plots for the all timescales of all
timescales of the monthly maxima in Brest are illustrated in Figure 10; they confirm the
suitability of the selected models.
Accordingly, the nonstationary GEV models have exhibited high improvements at the
interannual scales where the AIC scores have significantly decreased by introducing the
climate information into the parametrization of the model. Such consideration varies as a
function of the spectral components, it concerns all parameters for the smallest scale of ~1.5-
yr, both location and scale parameters for ~2-4-yr and only the location parameter for largest
scale of ~5-8-yr.
Then, the large-scale oscillations introduced for the implementation of GEV parameters depend
on the time scale for all sites exhibiting a high nonstationary behaviour of the small interannual
scales (~1.5-yr) which decreases at the large interannual scales (~5-8-yr) and get non-
significant at the interdecadal scales (~ 12-16-yr).
*Table 4 AIC test results for the distribution models of the extreme surges using the stationary*
*(GEV0) and the nonstationary (GEV1-3) models combined with climate oscillations indices. The*
*stationary (GEV0) and nonstationary GEV (GEV1, GEV2 and GEV3) models are illustrated for each*
*time scale and each site. The lowest AIC values for each case are marked by grey colour.*

The use of the time-varying GEV parameters at the interannual scales ($\sim$ 1.5-yr and $\sim$ 2-4-yr)
exhibits the relationship between the mode and the standard deviation of the GEV distributions
associated with the location and the scale parameters, respectively.
The different implications of both parameters for estimating the interannual extreme surges
reveal cyclic variations and timescale modulations related to the large-scale climate
oscillations. As documented in the previous works (e.g., Menendez et al., 2009; Masina and
Lamberti., 2013), the location and the scale parameters used for improving the nonstationary
estimation of the extreme water levels highlight a series of annual and semi-annual evolutions.
They have reported that the seasonal cycles of the location parameter are related to tow maxima
of water levels, in early March and September produced during equinoctial spring tides, while
the seasonal cycles of the scale parameter are associated to an increase of storms during wintry
episodes. Here, we focus on the stochastic signal of surges at scales larger than one year. The
SLP and the ZW frequencies, introduced in the location and the scale parameters of
nonstationary GEV models, determine an enhancement in the prediction of the interannual
scales.
The shape parameter, implied for the estimation of the ~1.5-yr extreme surges, derives from its
determination of the upper tail distribution behaviour.  The time-varying shape parameter uses
the ~1.5-yr SLP exhibiting altering negative and positive oscillations.
Despite its critical significance, the shape GEV parameter has revealed its relationships with
basin attributes in hydrological applications and regional flood frequency analysis (e.g., Tyralis
et al., 2019). The dependence of the shape parameter on the climate oscillations has been
demonstrated in several extreme frameworks related to hydrological and oceanographic
applications (e.g., Menendez et al., 2009; Masina et al., 2013; Turki et al., 2020). Regarding
the stationarity of the surge timescale, the ~12-16-yr window sliding matches have been
quantified in the previous part exhibiting a substantial cyclic variability consequence of an
altering periods of positive and negative correlations. The modelling of the interdecadal
extreme surges involves a stationary behavior of the ~12-16-yr.
Th stationary behavior of low frequencies has been outlined by Zampieri et al. (2017). They
have demonstrated a stationary trend of the SST anomalies associated with the AMO over the
Euro-Atlantic region. According to their works, the low-frequency variability of the European
Climate is influenced by the AMO shift induced by the phase opposition between the negative
NAO distribution and the Atlantic patterns.
Here, the effects of AMO on ~12-16-yr of extreme surges have been largely observed in Figure
9 for the longer timeseries Brest where the lower frequencies could be easily identified.

At this timescale, the AIC values given by the different GEV models are pretty close and the
difference between the distributions are not statistically significant.  The stationary behavior of
~12-16-yr surges should be more investigated from additional applications in light of the
available sea level measurements covering a long period of time, a relevant parameter to
characterize the uncertainties in extreme value statistical modeling of flood hazards.
*Figure 9  Wavelet details of monthly extreme surges (black lines), at the interannual (~ 1.5-yr , ~2-4-*
*yr and ~5-8-yr) and interdecadal (~12-16-yr) time scales for all sites (Brest, Cherbourg, Dunkirk, Dover*
*and Weymouth), correlated to the spectral component of climate oscillations associated to the*
*different indices  SLP, ZW, NAO and AMO (grey line).Only the connection maximizing the correlation*
*coefficient between a selected climate index and the component of surges (from interannual to the*
*interdecadal timescales) is presented (the normalized values have been calculated to superpose both*
*signals).*

*Figure 10 (a). The quantile plot between observed and modelled extreme surges by the use of the best*
*GEV models, at different time scales, case of Brest. (b). The Return level of extreme surges estimated*
*for Brest using the best GEV models. The 95% confidence interval is presented with the dashed black*
*line. (c) Fifty-year return level of monthly values using the original data (grey circles) and the best*
*nonstationary GEV model at Brest (solid black line). The lower and the upper limits of the 95%*
*confidence interval calculated using the delta method (dashed black line). The associated confidence*
*area is plotted with grey shaded area.*

The return levels of the multiscale extreme surges, provided by the best GEV models (Table
3), have been simulated. The example of Brest is illustrated in Figure 10.b for the interannual
(nonstationary GEV models) and the interdecadal (GEV stationary model) scales. The 95%
confidence interval is also plotted in this graph through a dashed black line. Accordingly, the
use of GEV distribution with time-dependent parameters for each timescale should improve
the evaluation of the return values and reduce the uncertainty of the quantile estimates.
Similar works have been carried out by Wahls and Chambers (2016) to investigate the
multidecadal variations in extreme sea levels with the large-scale climate variability. By the
use of climate indices on nearby atmospheric/oceanic variables (winds, pressure, sea surface
temperature) as covariates in a quasi-nonstationary extreme value analysis, the range of change
in the 100-year return water levels has been significantly reduced over time, turning a
nonstationary process into a stationary one.

As suggested by Wong (2018), including a wider range of physical process information and
considering nonstationary behaviour can better enable modelling efforts to inform coastal risk
management. In his work, he has developed a new approach to integrate stationary and
nonstationary statistical models and demonstrated that the choice of covariate timeseries should
affect the projected flood hazards. By developing a nonstationary storm surge statistical model
with the use of multiple covariate timeseries (global mean temperature, sea level, the North
Atlantic Oscillation index and time) in Norfolk and Virginia, he has shown that a storm surge
model raises the projected 100-year storm surge return level by up to 23 cm relative to a
stationary model or one that employs a single covariate timeseries.
This study has expanded the previous works of Turki et al. (2019; 2020) upon a new approach
combining spectral and probabilistic methods to integrate multiple streams of information
related to climate teleconnections. Indeed, each timescale has been simulated separately with
the nonstationary GEV models and expressed as a function of the most suitable climate index
improving its fitting. The estimation of the total signal of surges should be determined by
combining the developed nonstationary GEV models used for the different timescales.
These results should support the hypothesis introduced at the beginning of the present work
suggesting that: (i) the extreme surges should depend on different timescales; (ii) each
timescale should be related to a specific large-scale oscillation.
The finding is in agreement with the previous works of Lee et al. (2017) and Wang et al. (2018)
highlighting the importance of a careful consideration when complex physical mechanisms of
different climate indices are included into model structures for estimating extreme surges.
Indeed, this work provides a guidance on incorporating nonstationary processes of large-scale
oscillations to different spectral components informed by the wavelet techniques, the Bayesian
approaches and the GEV model probabilities.

The primary contribution of the present research is to present a new approach for: (1) investigating the multi-timescale variability of the nonstationary extreme surges; (2) identifying their multi-connection with climate oscillations according to the timescale and (3) resolve in part the problems of uncertainty of most appropriate climate to use as covariate for GEV models at each timescale. However, additional models (e.g. significance tests and sensitivity analyses and modelling uncertainties) and application sites (e.g. Mediterranean and pacific ones controlled by other climate oscillations) are required to expand the developed approach.

Also, generating a final robust stochastic model useful for projecting storm surge return levels and assessing the flood risk management requires further efforts to build on the potentially advantageous approach presented here by integrating the GEV models associated with the different timescales through the use of mathematical methods.

## 6. Summary and Concluding remarks

The dynamics of extreme surges together with the large-scale climate oscillations have been investigated by the use of hybrid methodological approach combining spectral analyses and nonstationary GEV models. Results have demonstrated that the interannual variability of extreme surges (~ 1.5-yr, ~2-4-yr and 5-8-yr) is around 25% for the French coasts and higher than 32% for the Britannic coasts; the interdecadal variability (~12-16-yr) varies between 2% and 4%. The fluctuations of extreme surges at ~1.5-yr are differently manifested between the different sites of the English Channel exhibiting a local variability of surges induced by the

effects of meteorological and oceanographic forces including changes in atmospheric pressures
and wind velocities in shallow water areas. Similar fluctuations have been observed at larger
scales of the interannual and the interdecadal variability. Changes in extreme surges (~1.5-yr,
~2-4-yr, ~5-8-yr and ~12-16-yr) have been proven to be significantly linked to atmospheric
oscillations (SLP, ZW, NAO and AMO, respectively) according to the timescale with a
nonlinear interaction between different oscillations at the same scale. This exhibits the complex
physical mechanisms of the global atmospheric circulation acting as a regulator and controlling
the local variability of extreme surges at different timescales. The connections between the
multiscale extreme surges and the internal modes of climate oscillations have been explored to
improve the estimation of extreme values by the use of nonstationary GEV models. The
simulated extreme surges have highlighted that introducing the climate oscillations for the
implementation of GEV parameters depends on the timescale for all sites; a high nonstationary
behaviour of the small interannual scales (~1.5-yr) decreases at the larger scales (~5-8-yr) and
seems to be non-significant at the interdecadal scales (~ 12-16-yr).
The conclusion of this research suggests that the physical mechanisms driven by the
atmospheric circulation, including the Gulf Stream gradients, play a key role in coastal extreme
surges. Establishing a strong connection of the large-scale climate oscillations with extreme
surges and flooding risks improve the estimation of the return levels.
This finding presents a handful of a new approach potentially useful as a first step forward for
(1) understanding the physical relation of downscaling from the global climate patterns to the
local extreme surges; (2) inferring the future projections of sea level change and extreme
events. Future work should build on this new approach to (1) improve the stochastic modelling
of the multi-timescale extreme surges; (2) incorporate others climate mechanisms known to be
important at local and regional scales for specific applications; (3) generate a robust tool for
the storm surge projections and the flood risk assessments based on the different timescale
models connected to their specific climate drivers.
**Acknowledgments**
The research programs 'RICOCHET', 'RAIV COT', 'REVE COT' and CNES-TOSCA
'COTEST' are acknowledged for finding this research. elated to the future mission of Surface
Water and Ocean Topography (SWOT). We also acknowledge the National Navy
Hydrographic Service (SHOM), the British Oceanographic Data Center and the National
Center for Environmental Prediction for providing sea level and atmospheric data.

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

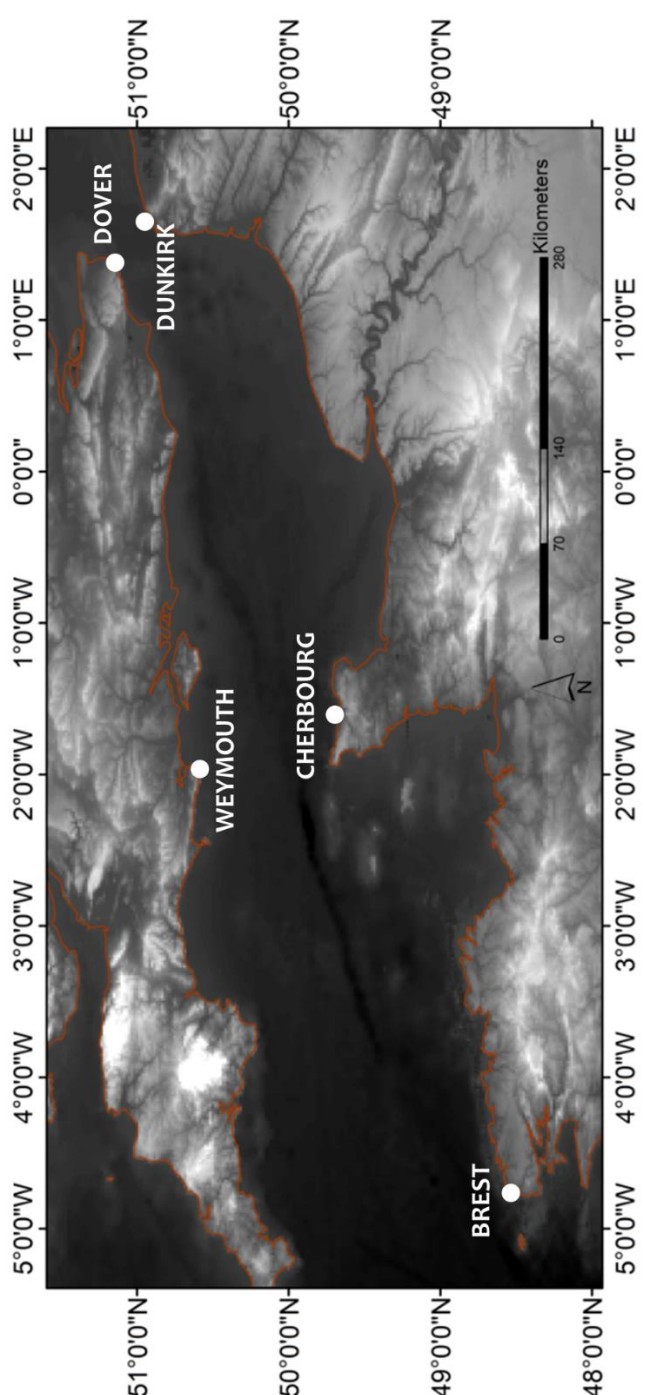



*Figure 1 Geographical location of the study area and the different tide gauges along the English*

*Channel coasts: Brest, Cherbourg, Dunkirk (NW France); Dover and Weymouth (SW UK).*



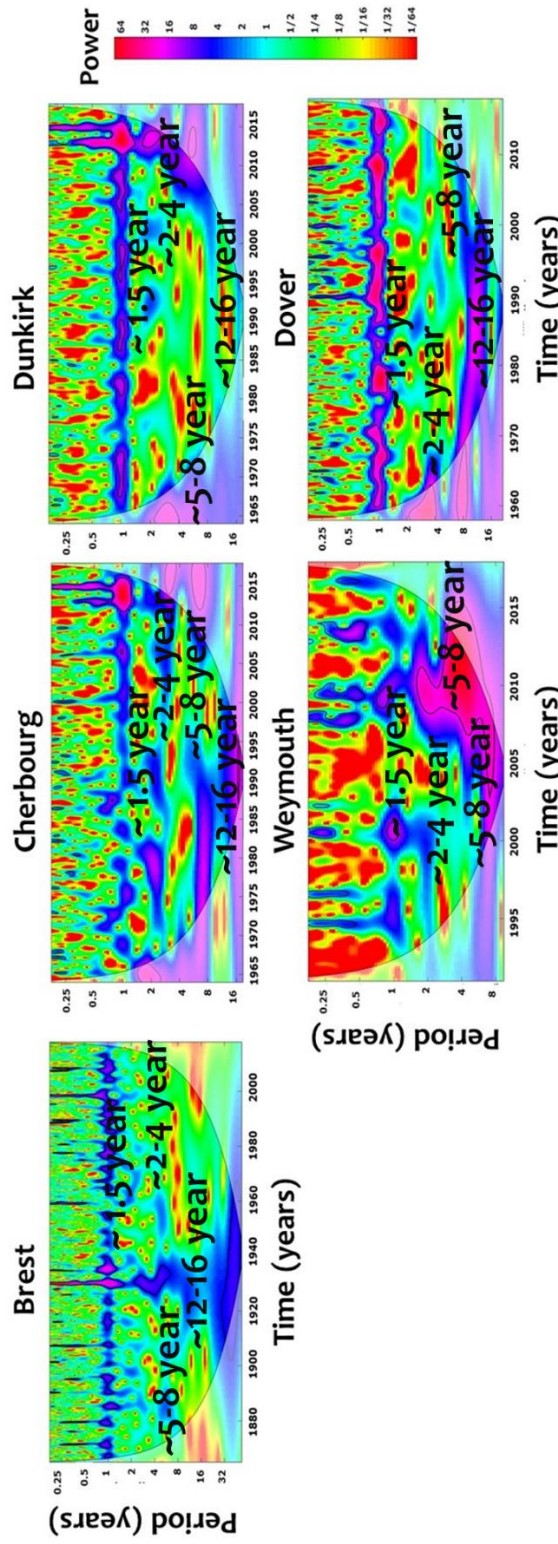


*Figure 2. CWT of monthly maxima of surges in Brest, Cherbourg, Dunkirk, Dover and*

*Weymouth.*



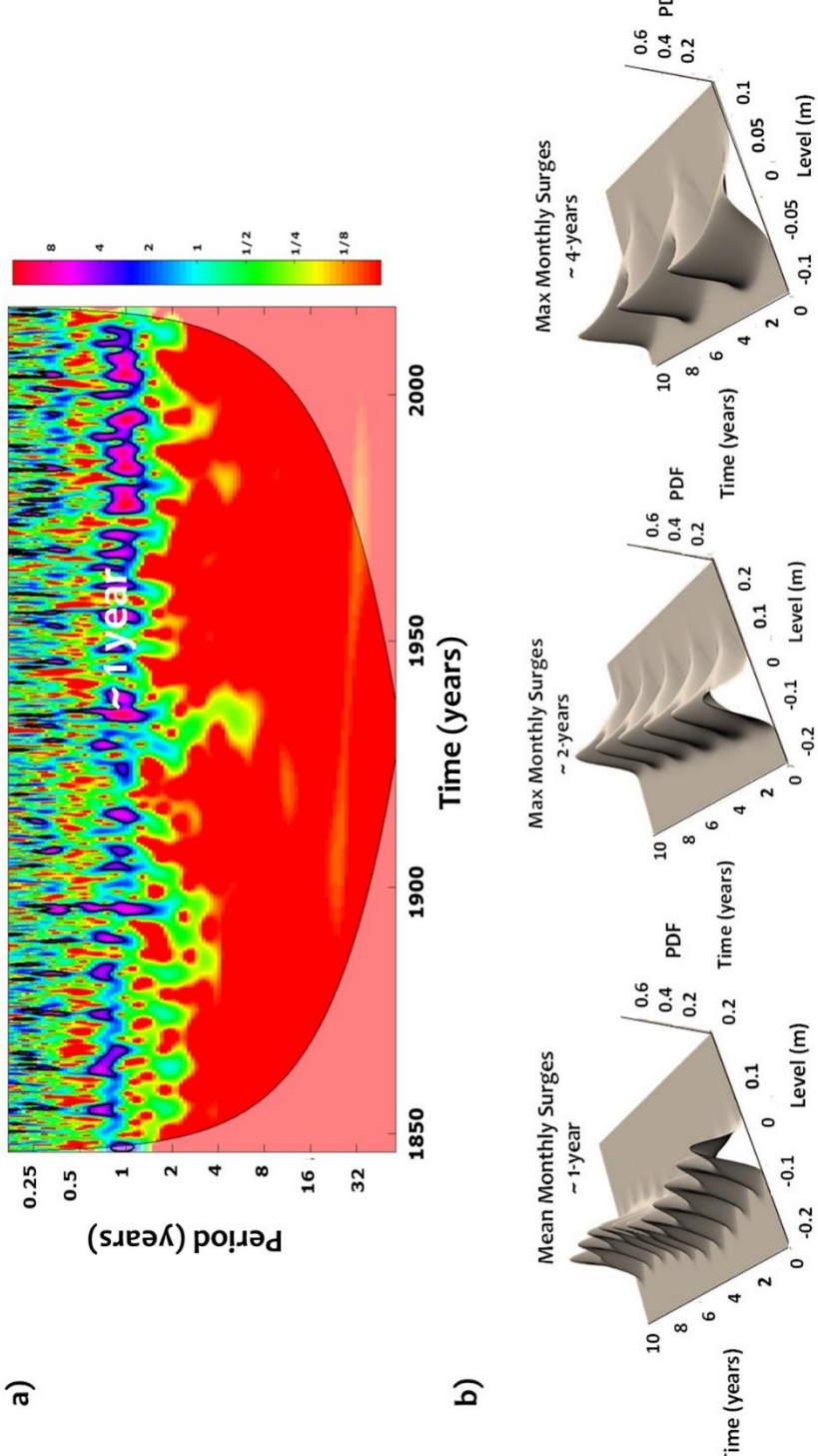


*Figure 3. Multiscale variability of the monthly mean and maximum surges in Brest. (a)*
*CWT of monthly mean surges; (b) Interannual variability of monthly and extreme surges*


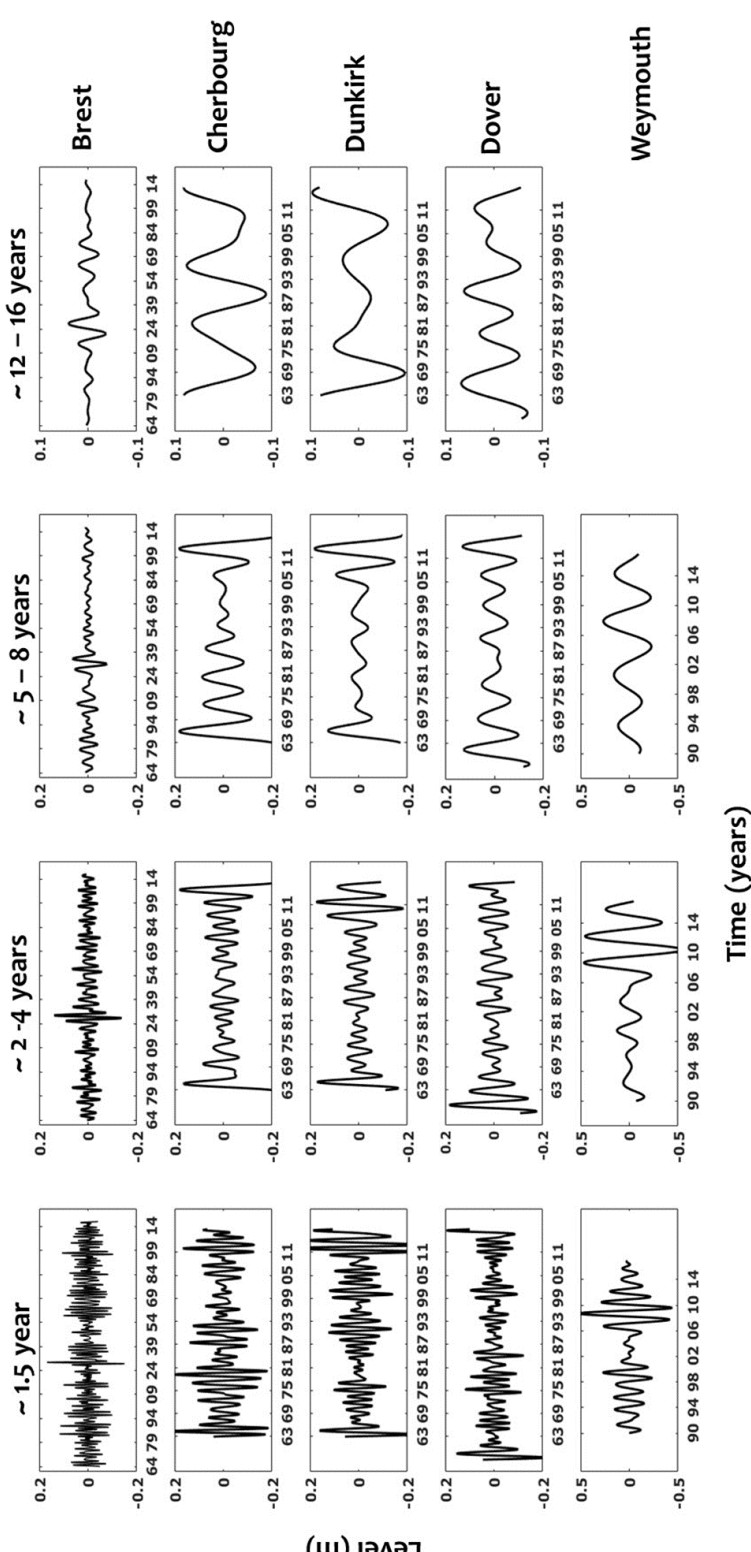


*Figure 4 Wavelet details (components) resulting from the multiresolution analysis of surges*
*at the interannual (~ 1.5-yr , ~2-4-yr and ~5-8-yr) and interdecadal (~12-16-yr) time scales*
*for all sites (Brest, Cherbourg, Dunkirk, Dover and Weymouth).*

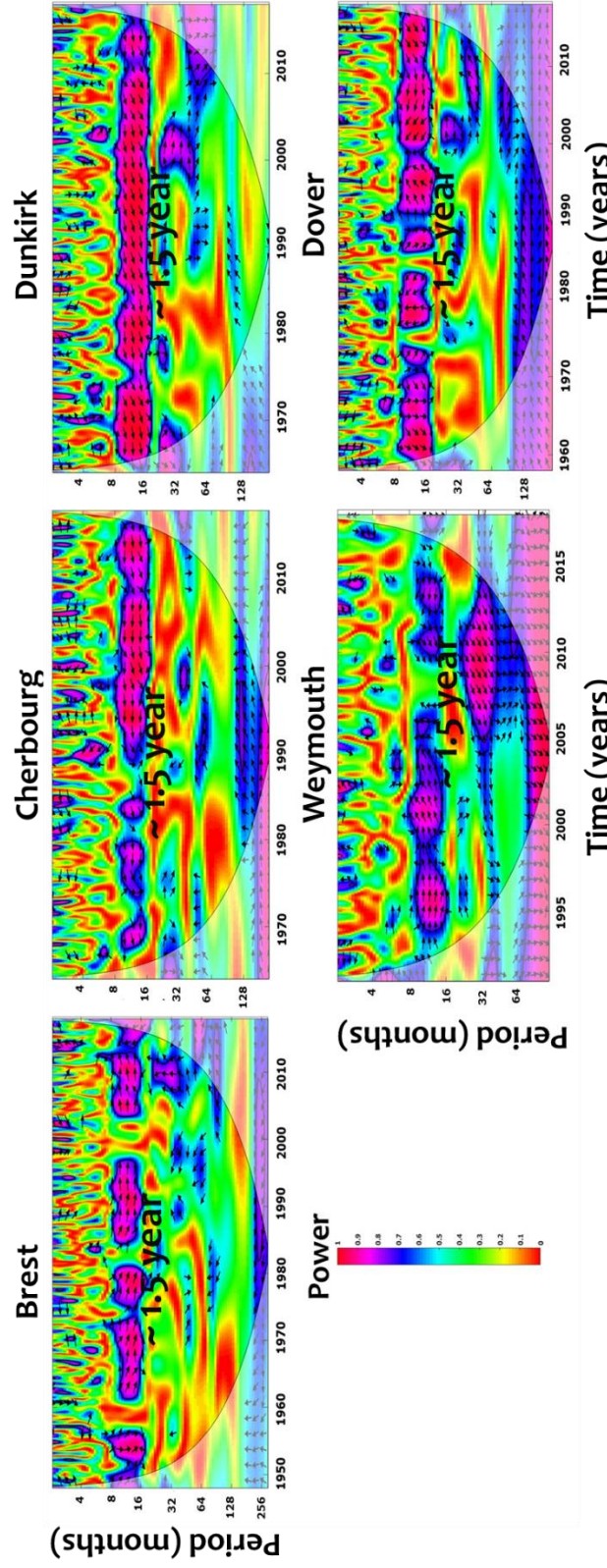


*Figure 5. Cross-wavelet correlations between monthly extrema of surges and Sea Level*
*Pressure (SLP).*


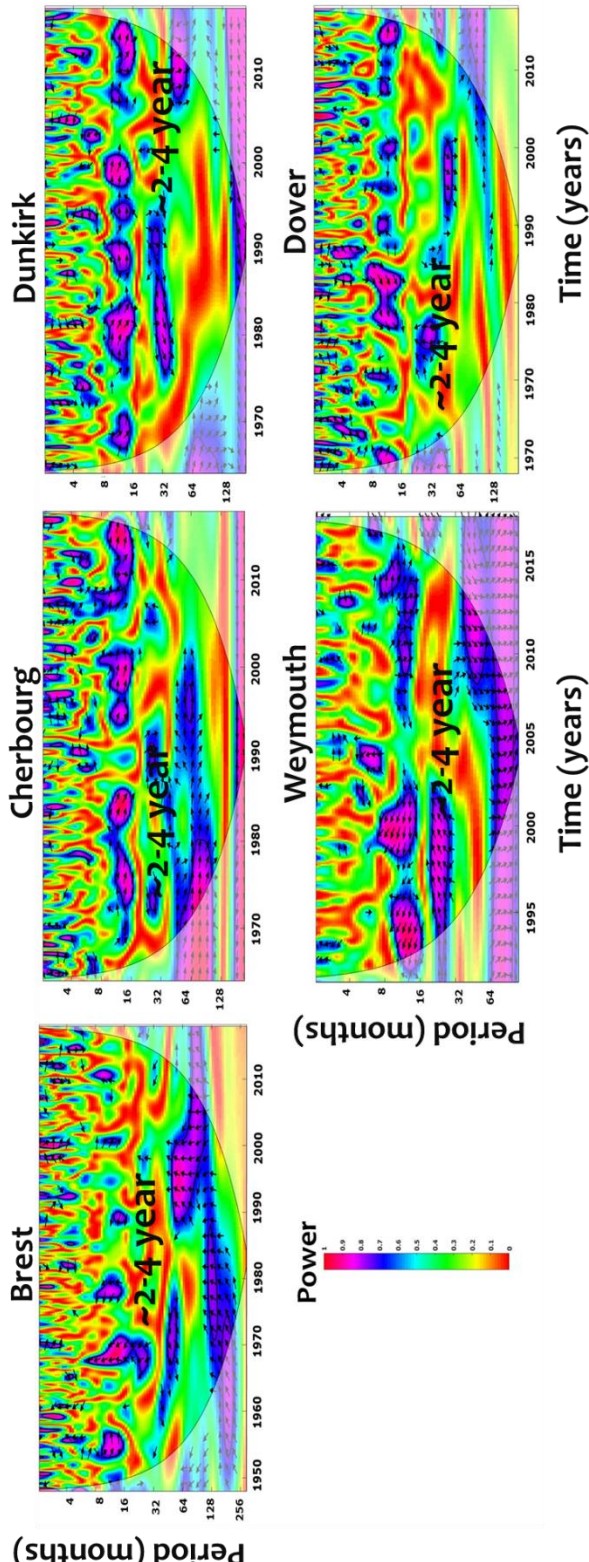


*Figure 6. Cross-wavelet correlations between monthly extrema of surges and Zonal Wind*

*(ZW).*


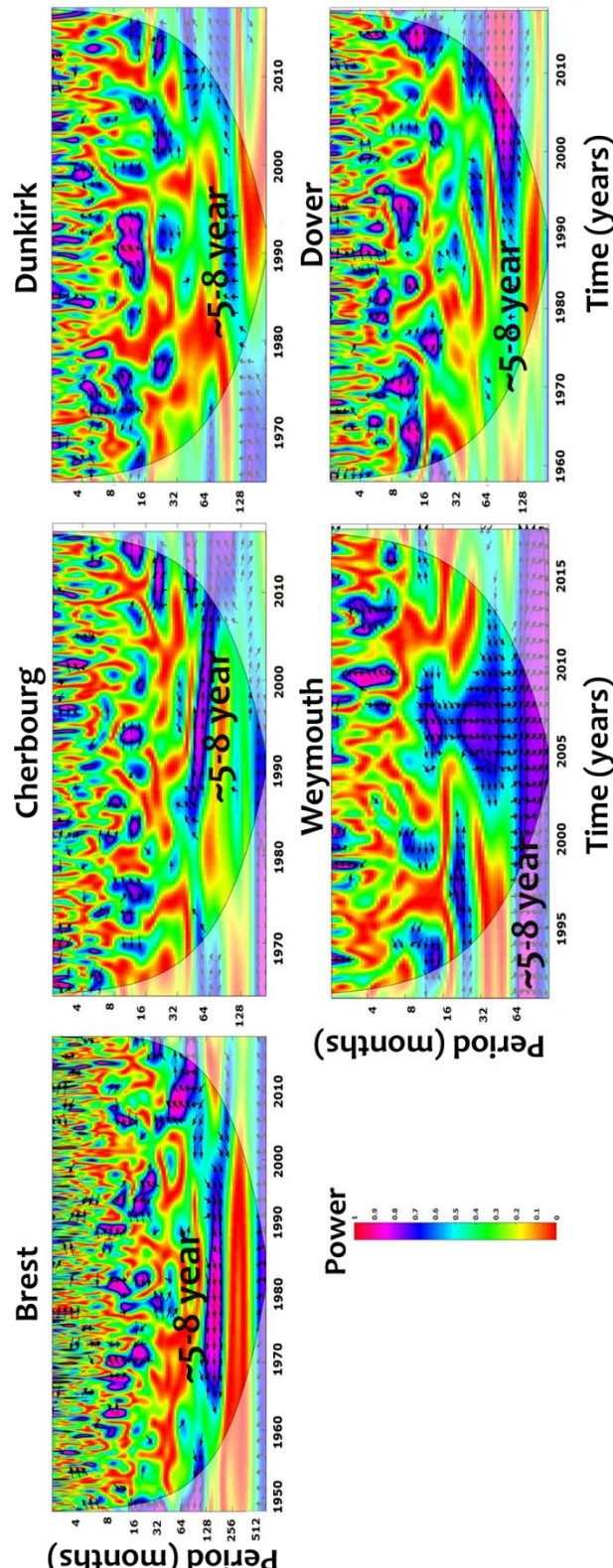


*Figure 7. Cross-wavelet correlations between monthly extrema of surges and North Atlantic*

*Oscillation (NAO).*


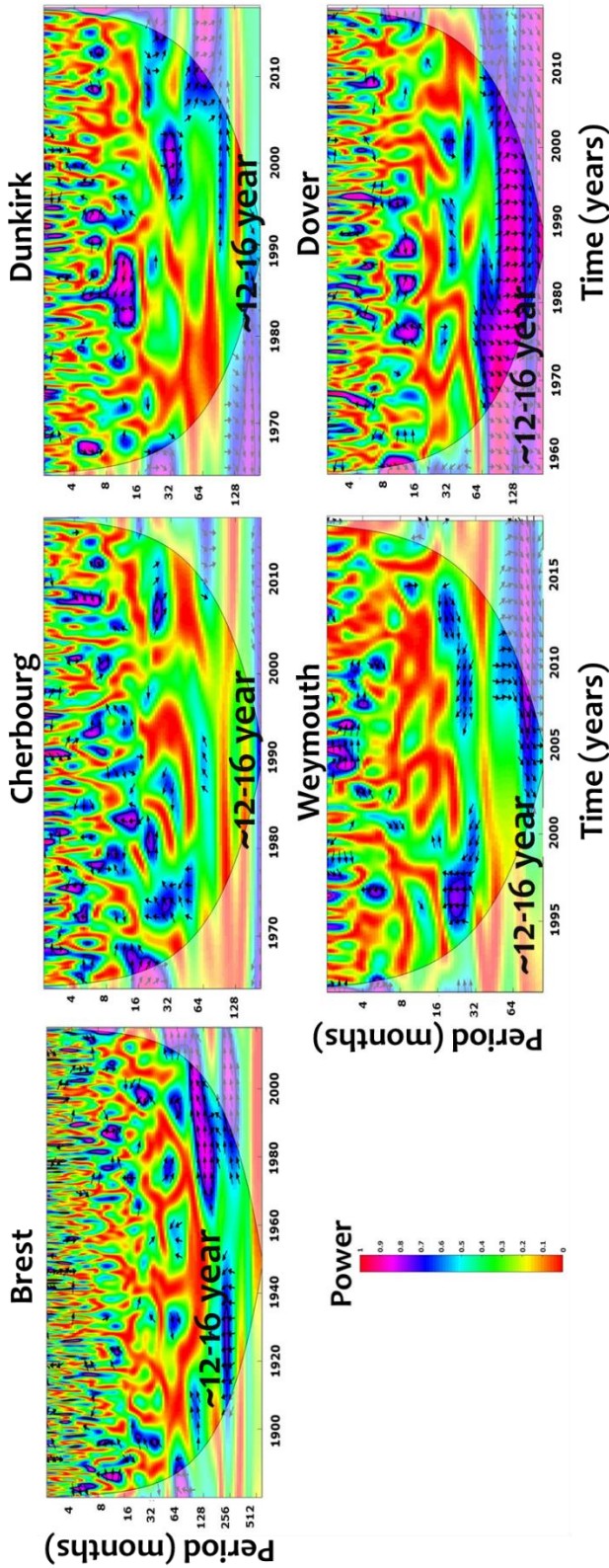


*Figure 8. Cross-wavelet correlations between monthly extrema of surges and Atlantic*
*Multidecadal Oscillation (AMO).*

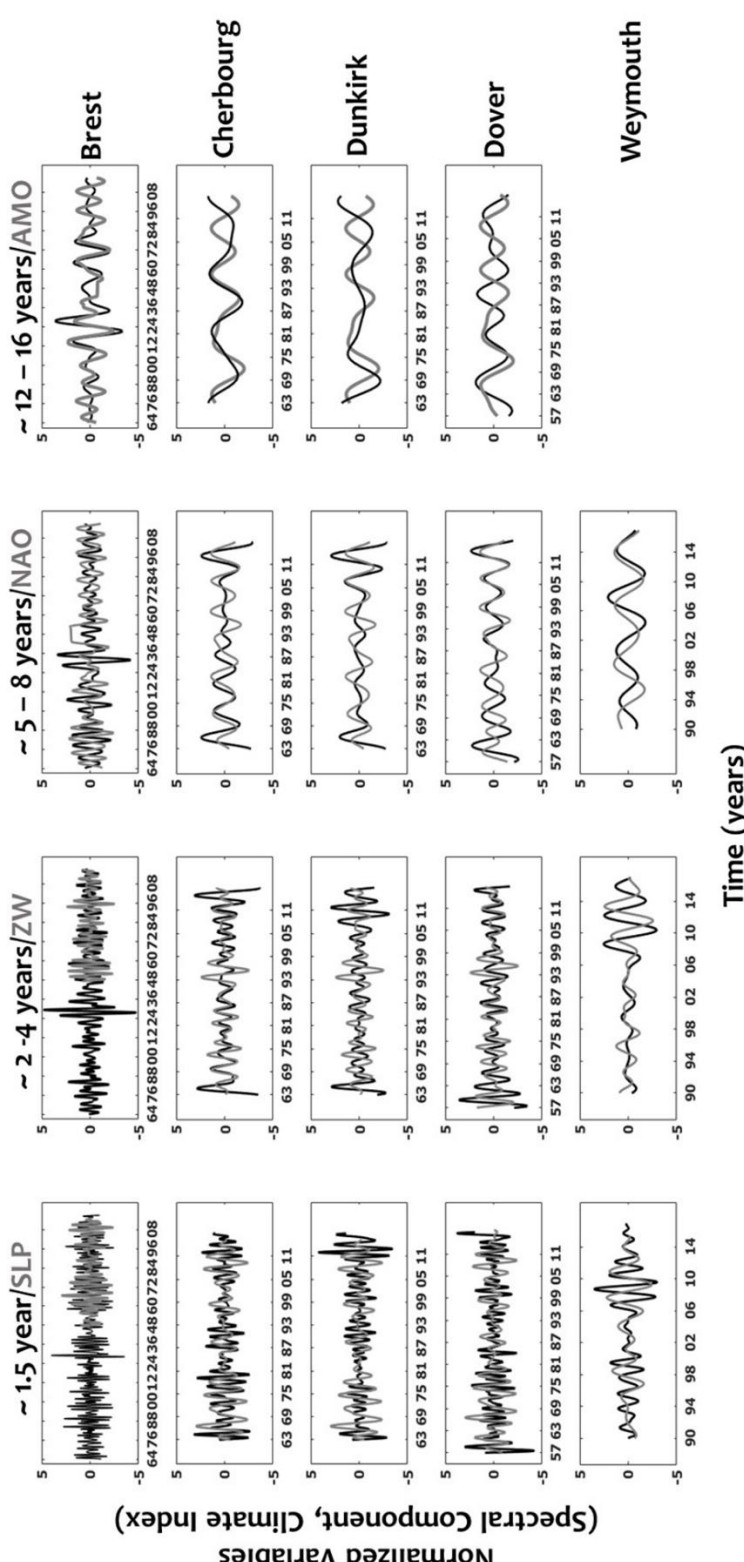



*Figure 9  Wavelet details of monthly extreme surges (black lines), at the interannual (~ 1.5-yr , ~2-4-*
*yr and ~5-8-yr) and interdecadal (~12-16-yr) time scales for all sites (Brest, Cherbourg, Dunkirk, Dover*
*and Weymouth), correlated to the spectral component of climate oscillations associated to the*
*different indices  SLP, ZW, NAO and AMO (grey line).Only the connection maximizing the correlation*
*coefficient between a selected climate index and the component of surges (from interannual to the*
*interdecadal timescales) is presented (the normalized values have been calculated to superpose both*
*signals).*


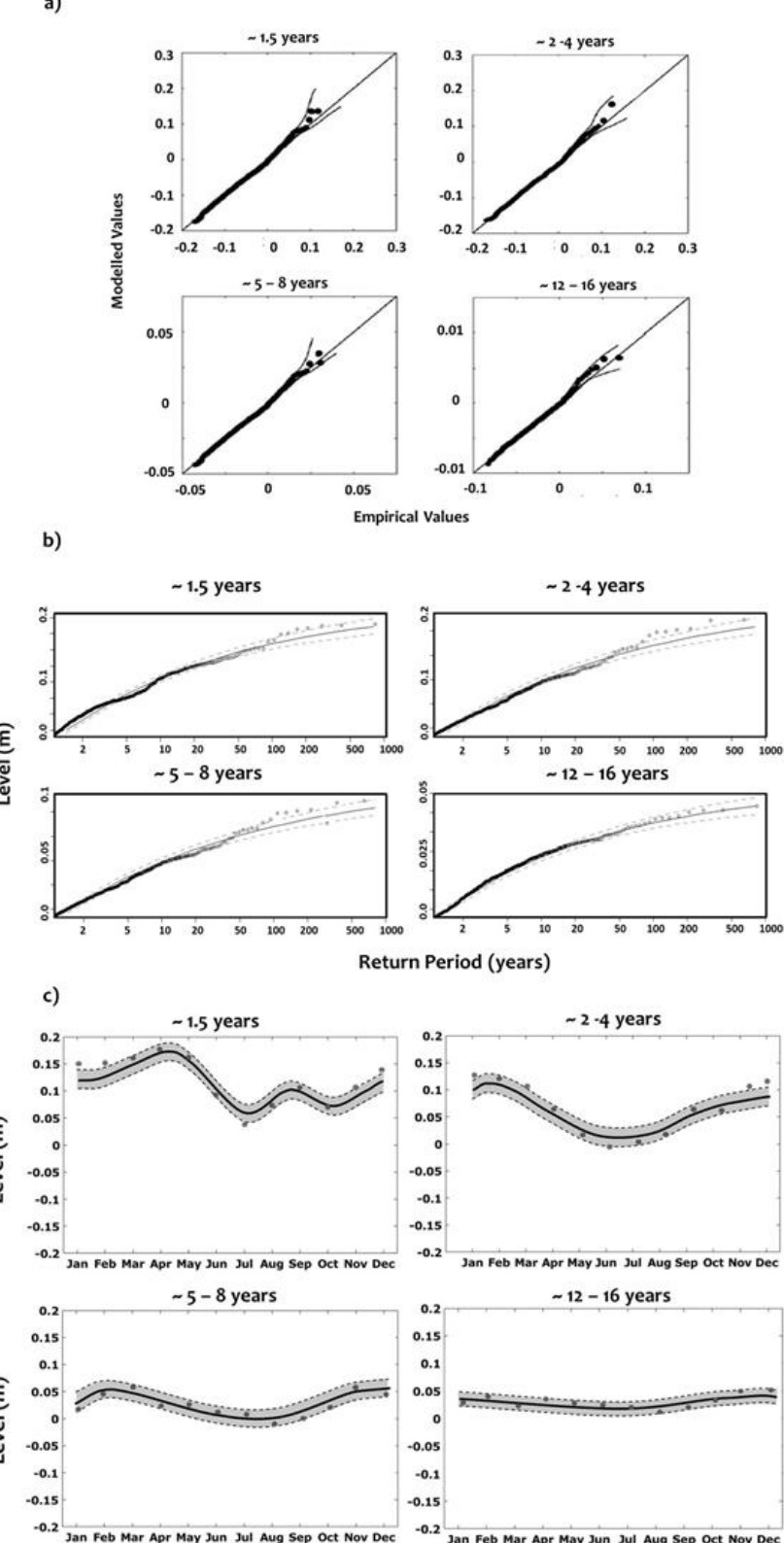

*Table 1. The explained variance expressed as percentage of total variance of monthly extreme surges for all sites.*

|  | ~ 1.5-yr | ~ 2-4-yr | ~ 5-8-yr | ~ 12-16-yr |
|---|---|---|---|---|
| **Brest** | **12.5%** | **7.5%** | **4.5%** | **1.9%** |
| **Cherbourg** | **14.8%** | **8.7%** | **5.2%** | **2.7%** |
| **Dunkirk** | **15.2%** | **8.6%** | **5.6%** | **3.2%** |
| **Dover** | **16.7%** | **9.9%** | **6.2%** | **3.9 %** |
| **Weymouth** | **16.5%** | **10.2%** | **7.9%** | |

*Table 2. The mean explained variance expressed as percentage of total variance provided by the wavelet coherence between the extreme surges and the climate Oscillations (SLP, ZW, NAO, AMO)*

|  | ~ 1.5-yr | ~ 2-4-yr | ~ 5-8-yr | ~ 12-16-yr |
|---|---|---|---|---|
| **SLP** | **75%** | **12%** | **15%** | **10%** |
| **ZW** | **10%** | **65%** | **12%** | **10%** |
| **NAO** | **6%** | **5%** | **60%** | **10%** |
| **AMO** | **0.2%** | **0.1%** | **1%** | **55 %** |

*Table 3 Analysis of the statistical significance of the correlation between the spectral component of the extreme surges and the climate oscillation at each timescale for the different stations. The 95% Confidence Intervals from Bootstrap technique in Square Brackets. The most significant correlations are illustrated by the grey columns.*

| ~ 1.5-yr | SLP | ZW | NAO | AMO |
|---|---|---|---|---|
| Brest | [0.152, 0.174] | [0.145, 0.182] | [0.141, 0.178] | [0.138, 0.189] |
| Cherbourg | [0.161, 0.170] | [0.142, 0.179] | [0.142, 0.179] | [0.135, 0.180] |
| Dunkirk | [0.160, 0.168] | [0.150, 0.185] | [0.150, 0.185] | [0.135, 0.183] |
| Dover | [0.158, 0.165] | [0.161, 0.180] | [0.161, 0.180] | [0.133, 0.180] |
| Weymouth | [0.421, 0.429] | [0.411, 0.450] | [0.381, 0.299] | [0.375, 0.281] |

| ~ 2-4-yr | | | | |
|---|---|---|---|---|
| Brest | [0.145, 0.164] | [0.149, 0.158] | [0.141, 0.179] | [0.138, 0.183] |
| Cherbourg | [0.160, 0.175] | [0.188, 0.196] | [0.161, 0.179] | [0.158, 0.182] |
| Dunkirk | [0.145, 0.158] | [0.180, 0.185] | [0.145, 0.164] | [0.140, 0.169] |
| Dover | [0.148, 0.163] | [0.192, 0.198] | [0.145, 0.168] | [0.143, 0.175] |
| Weymouth | [0.412, 0.420] | [0.420, 0.430] | [0.410, 0.425] | [0.410, 0.428] |

| ~ 5-8-yr | | | | |
|---|---|---|---|---|
| Brest | [0.075, 0.090] | [0.073, 0.092] | [0.085, 0.089] | [0.070, 0.096] |
| Cherbourg | [0.190, 0.198] | [0.185, 0.198] | [0.191, 0.196] | [0.181, 0.198] |
| Dunkirk | [0.180, 0.188] | [0.177, 0.185] | [0.183, 0.187] | [0.175, 0.187] |
| Dover | [0.180, 0.195] | [0.180, 0.198] | [0.180, 0.184] | [0.176, 0.199] |
| Weymouth | [0.219, 0.222] | [0.218, 0.225] | [0.221, 0.226] | [0.216, 0.226] |

| ~ 12-16-yr | | | | |
|---|---|---|---|---|
| Brest | [0.033, 0.046] | [0.034, 0.045] | [0.035, 0.045] | [0.038, 0.041] |
| Cherbourg | [0.089, 0,099] | [0.090, 0,099] | [0.090, 0,097] | [0.091, 0,095] |
| Dunkirk | [0.087, 0.099] | [0.089, 0.098] | [0.090, 0.097] | [0.093, 0.096] |
| Dover | [0.078, 0.089] | [0.080, 0.088] | [0.080, 0.086] | [0.082, 0.085] |
| Weymouth | [0.250, 0.260] | [0.250, 0.259] | [0.250, 0.257] | |


*Table 4  AIC test results for the distribution models of the extreme surges using the stationary*
*(GEV0) and the nonstationary (GEV1-3) models combined with climate oscillations indices. The*
*stationary (GEV0) and nonstationary GEV (GEV1, GEV2 and GEV3) models are illustrated for each*
*time scale and each site. The lowest AIC values for each case are marked by grey colour.*


| ~ 1.5-yr | GEV0 | GEV1 | GEV2 | GEV3 |
|---|---|---|---|---|
| **Brest** | **-2997** | **-3009** | **-3015** | **-3050** |
| **Cherbourg** | **-1591** | **-1620** | **-1622** | **-1662** |
| **Dunkirk** | **-1406** | **-1410** | **-1415** | **-1430** |
| **Dover** | **-2186** | **-2190** | **-2195** | **-2200** |
| **Weymouth** | **-2180** | **-2192** | **-2198** | **-2214** |


¶

| ~ 2-4-yr | GEV0 | GEV1 | GEV2 | GEV3 |
|---|---|---|---|---|
| **Brest** | **-3015** | **-3018** | **-3025** | **-3020** |
| **Cherbourg** | **-1511** | **-1620** | **-1642** | **-1622** |
| **Dunkirk** | **-1414** | **-1417** | **-1434** | **-1420** |
| **Dover** | **-2180** | **-2183** | **-2195** | **-2187** |
| **Weymouth** | **-2179** | **-2181** | **-2220** | **-2211** |



| ~ 5-8-yr | GEV0 | GEV1 | GEV2 | GEV3 |
|---|---|---|---|---|
| **Brest** | **-1962** | **-1975** | **-1922** | **-1940** |
| **Cherbourg** | **-1827** | **-1937** | **-1878** | **-1870** |
| **Dunkirk** | **-1797** | **-1850** | **-1815** | **-1810** |
| **Dover** | **-2175** | **-2198** | **-2168** | **-2160** |
| **Weymouth** | **-2171** | **-2180** | **-2162** | **-2158** |



| ~ 12-16-yr | GEV0 | GEV1 | GEV2 | GEV3 |
|---|---|---|---|---|
| **Brest** | **-1997** | **-1980** | **-1922** | **-1940** |
| **Cherbourg** | **-1225** | **-1212** | **-1205** | **-1198** |
| **Dunkirk** | **-1398** | **-1381** | **-1367** | **-1351** |
| **Dover** | **-1377** | **-1363** | **-1360** | **-1343** |
| **Weymouth** | | | | |





