# Peer review of "A nonstationary analysis for investigating the multiscale variability of extreme surges: case of the English Channel coasts"

_Natural Hazards and Earth System Sciences, 2020_

## Referee Comment (RC1) · Anonymous Referee #1 · 2 Jun 2020

**Review on the paper "A nonstationary analysis for investigating the multiscale variability of extreme surges: case of the English Channel coasts"**

**General comments**

In the research manuscript, the authors present an analysis of time series of storm surges in five stations along the coast of the English Channel.

The study is two-fold : a first part is related to the analysis of the monthly extreme storm-surges signals, by using a multi-scale wavelet analysis in order to describe the

**Specific comments**

[Figure]

**2. Data**

It no very clear in the paper if storm surge or the sea level height data is used. Only the later is measured by the tide gauge, and thus a pre-processing step is required to filter out the tides and the sea-level rise. The description of this pre-processing seems to be missing in the paper. Another question is about the availability of the large-scale atmospheric circulations indices (NAO, AMO...) during the whole period covered by the tide gauges. In particular, the Brest station has measurements from 1850, so one could wonder if the indices are available from the date, and what could be the quality of such values. I think that the paper could benefit from some discussion on this point.

**3. Extreme value models**

The authors use the classical extreme value distribution (GEV) to model the monthly maxima of storm surges, making the distribution non-stationary by incorporating climate indices as covariates.

Although the presentation of the model is rather clear, the data on with the model is applied is not as clear to me: is it on the initial time series of storm surges or on the spectral components? On L200, it seems that the model is applied to each spectral component, but the justification of using a GEV distribution is then questionable since the component by itself is not extreme nor a maxima, and thus an extreme value distribution is not justified. Marginal distributions of the variable on which the GEV is fitted could give some insight on the adequacy of a GEV, in addition to the QQ plots of Figure 10.

I have another remark about model selection: the authors do not show the fitted parameters values nor the associated confidence intervals, but only the AIC values in Table 3. Such values would be necessary to address the fit and to discuss whether or not the influences of the indices are significant. The authors are only selecting the parameter that cannot be considered are stationary, but not the index that is relevant to explain the non-stationary.

**4. Multi-timescale variability of extreme surges**

The authors describe the results of the continuous wavelet transform (CWT) on the monthly maxima of storm surges to assess the non-stationary behaviour

**5. Large-scale climate North-Atlantic oscillations and their link to extreme surges in the English Channel**

This section is two-fold : first, exhibit the link between the indices and the monthly maxima of storm surges and fit the GEV distribution to the components of the storm surge. The authors look at the wavelet coherence to address this question and conclude that :

> Each timescale exhibits mainly strong links with its associated climate index (L313)

Such a conclusion seems rather obvious to me because the indices are constructed that way and is not sufficient to my point of view to do variable selection in the GEV model of the following section before fitting the model.

Although the lengthy discussion about the visual inspection is interesting and may worth it, a proper statistical method to select the variable should be preferred.

Once the model is fitted, the paper falls short : since there is no variable selection and no use of the fitted model, what is the fit used for? We only can see with some difficulties the return levels of each component for the Brest station, but with little extrapolation. As is, the relevance of using a GEV model is questionable.

**Technical remarks**

- Spacing is not uniform in the text, please consider proof-reading; - Parameters values are missing in Table 3; - a Âű on L805; - Brest is missing in Figure 1; On L150: *The observations which correspond to the hydrographic zero level are referenced to zero tide gauge (Figure 1)* but seems missing; - Figures 2-3-5-6-7-8 : the time-scales can hardly be seen, please choose another color or another representation or another

colormap. - Figure 2 color bar is not coherent with the other Figures; - Figures 4 and 9: the x-axis is not clear, at least 4-digit years should be provided. - Figure 10 a : The y-axis should be "Modeled values"; - Figure 10 b: the figure can hardly be read, please provide a larger version or a better quality (e.g. SVG)
* * *

---

## Referee Comment (RC2) · Anonymous Referee #2 · 2 Jul 2020

The manuscript by Turki and co-authors addresses an important issue for the modeling of exceedance probability of extreme surges namely accounting for the dependence with climate patterns. The authors present an approach relying on wavelet analysis to investigate the correlation between the extreme surges and four climate oscillations (North Atlantic Oscillation, and Atlantic Multidecadal Oscillation, and the ones related to Sea-Level Pressure and Zonal Wind) at multiple time scales, ∼1.5-years, ∼2-4-years, and ∼5-8-years and 12-16-years. On this basis, they perform nonstationary extreme value analysis using the English Channel coasts as application cases and show the added-value for accounting for these multiscale processes when deriving the return periods.

Main comment

The manuscript is well organized and the presentation is clear. Yet, several aspects should be clarified and further elaborated before publication (state of the art, details of the implementation, discussion regarding the assumptions). Therefore, I recommend additional corrections by incorporating, if possible, the following recommendations.

Specific comments

1. State of the art.

Some key references about the link between extreme surges and climate variables should be added to the bibliography, namely: ** Marcos, M., Calafat, F.M., Berihuete, Á., Dangendorf, S., 2015. Long‐term variations in global sea level extremes. J. Geophys. Res. 120(12), 8115-8134. ** Marcos, M.; Woodworth, P.L., 2017. Spatiotemporal changes in extreme sea levels along the coasts of the North Atlantic and the Gulf of Mexico. J. Geophys. Res., 122(9), 7031-7048. ** Méndez, F. J., Menéndez, M., Luceño, A., Losada, I. J., 2007. Analyzing monthly extreme sea levels with a time-dependent GEV model. Journal of Atmospheric and Oceanic Technology 24(5), 894-911. ** Wahl, T., Chambers, D.P., 2015. Evidence for multidecadal variability in US extreme sea level records. J. Geophys. Res. 120, 1527–1544. ** Wahl, T.; Chambers, D.P., 2016. Climate controls multidecadal variability in US extreme sea level records. J. Geophys. Res. 121(2), 1274-1290.

My second concern relates to the differences of the present work with the recently published one, namely Turki et al. (2020). As far as I understood, the time scale 12-16-years and the British part of the Channel coasts were not tackled in this published work, but it would be useful to situate in more details the present study with respect to Turki et al. (2020), for instance in the introduction.

2. Details on the implementation.

The authors focus on extreme surges. To do so, the raw data of tide gauges should be

pre-processed by accounting for the tide. Could the authors provide more details on how they proceeded? What type of tide data did they used?

Similarly, the authors used climate indices provided by the NCEP-NCAR Reanalysis. Could the authors provide the web link where they downloaded the data for the climate indices? Besides, the authors mentioned climate oscillations using SLP and Zonal Wind. Are they directly available from NCEP-NCAR Reanalysis or are they derived from a pre-processing using EOF analysis for instance?

3. Model selection in the non-stationnary Extreme Value Analysis (EVA).

Integrating the climate drivers as covariates in EVA is a good idea, but the selection of the 'most appropriate' model deserves more discussion.

3.1. Adequacy of GEV.

It is not clear to me whether extreme value distributions are applied to each spectral component. If so, I wonder whether these variables are 'extreme', and whether GEV distribution is appropriate. Could the authors comment on that?

3.2. Variable selection.

Table 2 is used to select the most appropriate climate variables to be integrated in the EVA. Though informative and useful to support discussion, my concern is that this selection is mainly based on a correlation analysis (Figure 7 and following ones), and I wonder why the authors did not perform a variable selection for the GEV model directly; for instance using AIC or alternative selection criteria. See a discussion by Wong (2018)

3.3 Model selection.

Furthermore, the results for Brest in Table 3 may raise some questions: - For scale ~12-16 years, GEV0 does not seem to be the model that leads to the minimum AIC value (-1258 to be compared to -1980 for GEV1); - For scale ~ 2-4-yr, the AIC values

fro GEV1-3 are very close, which make very hard to identify with high confidence the most appropriate model. The authors should comment on that. See also Burnham and Anderson (2004) for further details.

Reference: Wong, T. E. (2018). An integration and assessment of multiple covariates of nonstationary storm surge statistical behavior by Bayesian model averaging. Advances in Statistical Climatology, Meteorology and Oceanography, 4(1/2), 53-63. Burnham, K. P. and Anderson, D. R.: Multimodel inference: understanding AIC and BIC in model selection, Sociolog. Meth. Res., 60, 261–304, 2004.

4. Correlation.

The authors analyze the significance of the correlation through a visual inspection of the results provided by wavelet spectral analysis. In lines 339-341, the authors mentioned that they are using a Monte-Carlo-based approach to identify the most statistically significant correlation: could the authors provide more details on the implementation. Is it a bootstrap-based approach? How do they analyse the changes of the correlation at the Monte-Carlo iterations? Could the authors provide additional results about this significance assessment?

5. Typo.

Line 70: "investigates" should be "investigate" Line 467: "covariable" should be covariate
* * *

---

## Author Comment (AC1) · 22 Aug 2020

**Dear Anonymous Reviewer**

We appreciate the time spent by the editor and the reviewer to assess the manuscript and we appreciate the constructive comments and suggestions proposed. We have taken into account all comments and we feel the manuscript has certainly benefited in terms of both clarity and content. Best Regards Imen Turki (also on behalf of the co-authors)

I present the answser to the comments above. I send it also in a pdf document.

Reviewer 1 : General Suggestions # Review on the paper "A nonstationary analysis for investigating the multiscale variability of extreme surges: case of the English Channel coasts" ## General comments In the research manuscript, the authors present an analysis of time series of storm surges in five stations along the coast of the English Channel. The study is two-fold: a first part is related to the analysis of the monthly extreme storm-surges signals, by using a multi-scale wavelet analysis in order to describe the

Answer to Reviewer 1 : Thank you for the different suggestions and comments useful for the improvement of the manuscript. The text of the manuscript has been checked by the authors and has been simplified to make easier the writing and the understanding of the different sections, in particular the discussion and the methodology. Also, some information has been moved from the methodological part to the results. Two illustrations related to the original hydrodynamic data (Figure 4 in the new version) and the morphological defects (Figure 2 in the new version). The answers will be addressed for each specific comment. ### 2. Data It no very clear in the paper if storm surge or the sea level height data is used. Only the later is measured by the tide gauge, and thus a pre-processing step is required to filter out the tides and the sea-level rise. The description of this pre-processing seems to be missing in the paper. Another question is about the availability of the large-scale atmospheric circulations indices (NAO, AMO...) during the whole period covered by the tide gauges. In particular, the Brest station has measurements from 1850, so one could wonder if the indices are available from the date, and what could be the quality of such values. I think that the paper could benefit from some discussion on this point. Answer Thank you for this suggestion. According to the determination of surges, a new part explaining this extraction has been added in the new version: part 3.1; According to the availability of the largescale atmospheric circulations indices (NAO, AMO...), they have been extracted from the NCEP-NCAR Reanalysis fields with the same period of the surges. a. Timeseries of surges Weymouth: 1991-2018 Brest: 1846-2018 Dunkirk:1964-2018 Dover: 1958-2018 Cherbourg: 1964-2018

**NHESSD**
b. Timeseries of the correlations Climate index - maximum surges The correlation has been carried depending on the length of the

AMO/BREST: 1880 – 2017 NAO /Brest : 1865-2017 SLP /Brest : 1948-2017 ZW /Brest : 1865-2017

AMO – NAO – SLP - ZW/ Cherbourg - Dunkirk : 1964-2017 AMO – NAO – SLP - ZW/Douvres : 1958-2017 AMO – NAO – SLP- ZW/Weymouth : 1991-2017

The methodological approach has benefit from this information.

Extraction of surges: A new part has been added in the manuscript (lines 191-213) 3. 1 Extraction of residual sea level: 'surges' The total sea-level height, resulting from the astronomical and the meteorological processes, exhibits a temporal non-stationarity which is explained by a combination of the effects of the long-term trends in the mean sea level, the modulation by the deterministic tidal component and the stochastic signal of surges, and the interactions between tides and surges. The occurrence of extreme sea levels is controlled by periods of high astronomically generated tides, in particular at inter-annual scales when two phenomena of precession cause systematic variation of high tides. The modulation of the tides contributes to the enhanced risk of coastal flooding. Therefore, the separation between tidal and non-tidal signals is an important task in any analysis of sea-level time-series. By the hypothesis of independence between the astronomical tides and the stochastic residual of surges, the nonlinear relationship between the tidal modulation and surges is not considered in the present analysis. Using the classical harmonic analysis, the tidal component has been modelled as the sum of a finite set of sinusoids at specific frequencies to determine the determinist phase/ amplitude of each sinusoid and predict the astronomical component of tides. In order to obtain a quantitative assessment of the non-tidal contribution in storminess changes, technical methods based on MATLAB t-tide package have been applied to the seal level measurements, demodulated from long-term components (e.g. mean sea level, vertical local movement ), for estimating year-by-year

**NHESSD**
tidal constituents. A year-by-year tidal simulation (Shaw and Tsimplis, 2010) has been applied to the sea-level time-series to determine the amplitude and the phase of tidal modulations using harmonic analysis fitted to 18.61-, 9.305-, 8.85-, and 4.425-year sinusoidal signals (Pugh, 1987). The radiational components have been also considered for the extraction of the stochastic component of surges (Williams et al., 2018). Detailed information related to Climate Oscillations A new part has been added in the manuscript (lines 183-189)

Monthly time-series of climate indices have been provided by the NCEP-NCAR Reanalysis fields (http://www.esrl.noaa.gov/psd/data/gridded/data.ncep.reanalysis.derived.html) until 2017. The different indices have been extracted during the same period of the sea-level observations at the four stations Cherbourg, Dunkirk, Dover and Weymouth. For the longest timeseries of Brest (1850 - 2018), the use of climate indices has been limited according to their initial date availability (AMO: 1880 – 2017; NAO: 1865-2017; SLP: 1948-2017; ZW: 1865-2017)

**3. Extreme value models The authors use the classical extreme value distribution (GEV) to model the monthly maxima of storm surges, making the distribution nonstationary by incorporating climate indices as covariates. Although the presentation of the model is rather clear, the data on with the model is applied is not as clear to me: is it on the initial time series of storm surges or on the spectral components? On L200, it seems that the model is applied to each spectral component, but the justification of using a GEV distribution is then questionable since the component by itself is not extreme nor a maxima, and thus an extreme value distribution is not justified. Marginal distributions of the variable on which the GEV is fitted could give some insight on the adequacy of a GEV, in addition to the QQ plots of Figure 10. I have another remark about model selection: the authors do not show the fitted parameters values nor the associated confidence intervals, but only the AIC values in Table 3. Such values would be necessary to address the fit and to discuss whether or not the influences of the indices are significant. The authors are only selecting the parameter that cannot be considered**
are stationary, but not the index that is relevant to explain the non-stationary.

Answer The authors try to add more clarifications related to the application of the GEV model. Indeed, the maxima of surges has been decomposed to different frequencies to which the model has been applied. This approach has been applied by Turki et al., 2019 ; 2020. The low and the high frequencies of the maxima highlight the different fluctuations of the signal and their multiscale variability. The prediction of each fluctuation has been investigated by the use of nonstationary GEV model with the incorporation of climate indices.

Some information related to the calculation of the return periods/levels and confidence intervals has been added 3.3 Stationary and Nonstationary extreme value model (lines 261-263) The non-stationary return levels, return-periods and the confidence intervals have been calculated by the use of a Bayesian inference models with the Maximum Likelihood Estimation. ### 4. Multi-timescale variability of extreme surges The authors describe the results of the continuous wavelet transform (CWT) on the monthly maxima of storm surges to assess the non-stationary behavior; Answer The authors have used the CWT to identify the spectrum of the extreme signal and its distribution in space (between station) and time (during the period of study); (Figure 2). The authors have shown also (Fig 3) that the low frequencies ( lower than 10 year) are clearly observed from the CWT of the extreme surges, and less identified from the mean surges. In this part, a quantification of the spectral components has been presented (Figure 4).

**5. Large-scale climate North-Atlantic oscillations and their link to extreme surges in the English Channel**

This section is two-fold : first, exhibit the link between the indices and the monthly maxima of storm surges and fit the GEV distribution to the components of the storm surge. The authors look at the wavelet coherence to address this question and conclude that: Each timescale exhibits mainly strong links with its associated climate index (L313) Such a conclusion seems rather obvious to me because the indices are constructed
that way and is not sufficient to my point of view to do variable selection in the GEV model of the following section before fitting the model.

Although the lengthy discussion about the visual inspection is interesting and may worth it, a proper statistical method to select the variable should be preferred. Once the model is fitted, the paper falls short : since there is no variable selection and no use of the fitted model, what is the fit used for? We only can see with some difficulties the return levels of each component for the Brest station, but with little extrapolation. As is, the relevance of using a GEV model is questionable. Answer Thank you for these suggestions. Indeed, according to the previous works, the effects of the climate patterns is important and should be considered for predicting the variability of extreme surges. In the present research, the novel approach that an excellent prediction of the total signal requires a good multi-timescale prediction, i.e a prediction of each spectral component (provided by the MODWT analysis) which is described by an appropriate climate index (the most appropriate one has been selected basing on the wavelet coherence and the monte carlo iterations for each component). Indeed, the non-linear interaction between the physical mechanisms of climate patterns is very complex and could not considered at the same time to predict the extreme surges. To investigate this interaction our hypothesis consists on developing a statistical model able to predict the spectral component with the incorporation of the most adequate climate information. The development of a full model useful for estimating the extreme surges needs the integration of the GEV models associated to the different timescales ( $\sim$  2-4 years;  $\sim$  5-8 years;  $\sim$  12-16 years) by the means of mathematical methods; which is the objective of the further works. The present work brings a novel hypothesis to resolve the complex effects of climate patterns on the local variability of surges. This step is very important to introduce a new model considering the different timescales of extreme surges.

More clarifications related to the selection of the most appropriate climate oscillation (lines 264-283) 2. 3. 4 Determination of the most appropriate climate oscillation connected to each timescale extreme surges for GEV models As suggested previ-
ously, the main hypothesis presented in this research is that effects of the physical mechanisms on the extreme surges varies according to the timescale and each scale should be related to a given climate oscillation. This hypothesis has been supported by two approaches: (1) a spectral approach based on the use of wavelet techniques (wavelet multiresolution and wavelet coherence as detailed in section 3.2) for optimizing the physical relationship between climate index and the extreme surges at each timescale; (2) a Bayesian approach has been used also for assessing extremes in a changing climate oscillation (NAO, SLP, ZW and AMO) at each timescale by making inferences from the Likelihood function. In our case where many parameters of GEV distribution should be optimized by including the most appropriate climate oscillation, Markov Chain Monte Carlo (MCMC) techniques have been implemented based on multiple simulations (the number of simulations varying as a function of the length of the timeseries; it is around 100.000 simulations). For generating the sequences of simulated values, we have applied the evbayes package within R software. By the use of this algorithm, a sequence of parameters with a normal distribution (a mean value equal to the previous value in the chain and a given variance). The most suitable climate oscillation maximizing the fitting between the observed and the simulated data is identified when a burn-in-period is reached.

Please also note the supplement to this comment: https://nhess.copernicus.org/preprints/nhess-2020-101/nhess-2020-101-AC1supplement.pdf

**NHESSD**
**Fig. 1.** Figure 1 Geographical location of the study area and the different tide gauges along the English Channel coasts: Brest, Cherbourg, Dunkirk (NW France); Dover and Weymouth (SW UK).

---

## Author Comment (AC2) · 22 Aug 2020

Dear Anonymous Reviewer We appreciate the time spent by the editor and the reviewer to assess the manuscript and we appreciate the constructive comments and suggestions proposed. We have taken into account all comments and we feel the manuscript has certainly benefited in terms of both clarity and content. Best Regards Imen Turki (also on behalf of the co-authors) I present the answser to the comments above. I send it also in a pdf document.

Answer to Reviewer 2 : Specific comments 1. State of the art. Some key references about the link between extreme surges and climate variables should be added to the

bibliography, namely: ** Marcos, M., Calafat, F.M., Berihuete, Á., Dangendorf, S., 2015. Long term variations in global sea level extremes. J. Geophys. Res. 120(12), 8115-8134.

** Marcos, M.; Woodworth, P.L., 2017. Spatiotemporal changes in extreme sea levels along the coasts of the North Atlantic and the Gulf of Mexico. J. Geophys. Res., 122(9), 7031-7048.

** Wahl, T., Chambers, D.P., 2015. Evidence for multidecadal variability in US extreme sea level records. J. Geophys. Res. 120, 1527–1544.

** Wahl, T.; Chambers,D.P., 2016. Climate controls multidecadal variability in US extreme sea level records. J. Geophys. Res. 121(2), 1274-1290. My second concern relates to the differences of the present work with the recently published one, namely Turki et al. (2020). As far as I understood, the time scale 12- 16-years and the British part of the Channel coasts were not tackled in this published work, but it would be useful to situate in more details the present study with respect to Turki et al. (2020), for instance in the introduction.

Thank you for the comments. The proposed references have been revised and added to the state of the art. The new funding proposed in this research, compared to the last work of Turki et al., 2020, has been better explained in the state of the art as suggested in line . . . . . . . . . . . . . . . . . . . References Proposed: 1. Introduction (lines 75 -90) Then, Marcos et al. (2015) have investigated the decadal and multidecadal changes in sea level extremes using long tide gauge records distributed worldwide. They have demonstrated that the intensity and the occurrence of the extreme sea levels vary on decadal scales in the most of the sites in relation with a common large-scale forcing. In the same way, the study of extreme sea levels along the coastal zones of the North Atlantic Ocean and the Gulf of Mexico has shown that the mean sea level should be considered as the major driver of extremes (Marcos and Woodworth 2017) since the intensity of extreme episodes increases at centennial time scales, together with multi-

[revised manuscript text omitted]

The authors focus on extreme surges. To do so, the raw data of tide gauges should be pre-processed by accounting for the tide. Could the authors provide more details on how they proceeded? What type of tide data did they used? Similarly, the authors used climate indices provided by the NCEP-NCAR Reanalysis. Could the authors provide the web link where they downloaded the data for the climate indices? Besides, the authors mentioned climate oscillations using SLP and Zonal Wind. Are they directly available from NCEP-NCAR Reanalysis or are they derived from a pre-processing using EOF analysis for instance? 3. Model selection in the non-stationnary Extreme Value Analysis (EVA). Integrating the climate drivers as covariates in EVA is a good idea, but the selection of the 'most appropriate' model deserves more discussion.

Regarding the extraction of extreme surges, more details related to the classical model used for calculating tides are provided in the new version. Also, the different climate indices have been better explained. The selection of the most appropriate GEV model has been achieved for each frequency component. The use of the climate information has been differently explained for the different spectral component. More explanations related to this part has been added in the new version (a new section in the methodological approach has been added) Extraction of surges: A new part has been added in the manuscript (lines 191-213) 3. 1 Extraction of residual sea level: 'surges' The total sea-level height, resulting from the astronomical and the meteorological processes, exhibits a temporal non-stationarity which is explained by a combination of the effects of the long-term trends in the mean sea level, the modulation by the deterministic tidal component and the stochastic signal of surges, and the interactions between tides and surges. The occurrence of extreme sea levels is controlled by periods of high astronomically generated tides, in particular at inter-annual scales when two phenomena of precession cause systematic variation of high tides. The modulation of the tides contributes to the enhanced risk of coastal flooding. Therefore, the separation between tidal and non-tidal signals is an important task in any analysis of sea-level time-series. By the hypothesis of independence between the astronomical tides and the stochastic residual of surges, the nonlinear relationship between the tidal modulation and surges is not considered in the present analysis. Using the classical harmonic analysis, the tidal component has been modelled as the sum of a finite set of sinusoids at specific frequencies to determine the determinist phase/ amplitude of each sinusoid and predict the astronomical component of tides. In order to obtain a quantitative assessment of the non-tidal contribution in storminess changes, technical methods based on MATLAB t-tide package have been applied to the seal level measurements, demodulated from long-term components (e.g. mean sea level, vertical local movement ), for estimating year-by-year tidal constituents. A year-by-year tidal simulation (Shaw and Tsimplis, 2010) has been applied to the sea-level time-series to determine the amplitude and the phase of tidal modulations using harmonic analysis fitted to 18.61-, 9.305-, 8.85-, and 4.425-year sinusoidal signals (Pugh, 1987). The radiational components have been also considered for the extraction of the stochastic component of surges (Williams et al., 2018). Detailed information related to Climate Oscillations A new part has been added in the manuscript (lines 183-189)

Monthly time-series of climate indices have been provided by the NCEP-NCAR Reanalysis fields (http://www.esrl.noaa.gov/psd/data/gridded/data.ncep.reanalysis.derived.html) until 2017. The different indices have been extracted during the same period of the sea-level observations at the four stations Cherbourg, Dunkirk, Dover and Weymouth. For the longest timeseries of Brest (1850 - 2018), the use of climate indices has been limited according to their initial date availability (AMO: 1880 − 2017; NAO: 1865-2017; SLP: 1948-2017; ZW: 1865-2017) Selection of the most appropriate climate oscillation (lines 264-283) 3. 3. 4 Determination of the most appropriate climate oscillation connected to each timescale extreme surges for GEV models As suggested previously, the main hypothesis presented in this research is that effects of the physical mechanisms on the extreme surges varies according to the timescale

and each scale should be related to a given climate oscillation. This hypothesis has been supported by two approaches: (1) a spectral approach based on the use of wavelet techniques (wavelet multiresolution and wavelet coherence as detailed in section 3.2) for optimizing the physical relationship between climate index and the extreme surges at each timescale; (2) a Bayesian approach has been used also for assessing extremes in a changing climate oscillation (NAO, SLP, ZW and AMO) at each timescale by making inferences from the Likelihood function. In our case where many parameters of GEV distribution should be optimized by including the most appropriate climate oscillation, Markov Chain Monte Carlo (MCMC) techniques have been implemented based on multiple simulations (the number of simulations varying as a function of the length of the timeseries; it is around 100.000 simulations). For generating the sequences of simulated values, we have applied the evbayes package within R software. By the use of this algorithm, a sequence of parameters with a normal distribution (a mean value equal to the previous value in the chain and a given variance). The most suitable climate oscillation maximizing the fitting between the observed and the simulated data is identified when a burn-in-period is reached. 3.1. Adequacy of GEV. It is not clear to me whether extreme value distributions are applied to each spectral component. If so, I wonder whether these variables are 'extreme', and whether GEV distribution is appropriate. Could the authors comment on that?

The monthly extreme surges have been calculated from hourly residual sea level. This signal has been decomposed by the MODWT to study separately the different components. Our hypothesis in the present research is the following: The variability of the local extreme surges should be explained by the global climate patterns described by a series of physical mechanisms associated to the climate indices. We have used the hypothesis that each spectral should be explained by a climate mechanism. Such hypothesis has been justified and validated by (1) the coherence diagrams (see also Table 2) where we have demonstrated that the effect of each climate index on the variability of extreme surges varies as a function of the spectral component and (2) Bayesian approaches applied to each spectral component to select the most appropriate climate index. This analysis has shown a strong coherence with the first validation (1).

This suggestion has been considered in the new version by incorporating a new section 3. 4/

Also more clarifications in the section 5.2 Nonstationary modelling of extreme surges (lines 538-548)

The connections between the climate oscillations and the monthly maxima at the different timescales (Figure 9)., presented previously (section 5.1), have been explored as a first hypothesis for the implementation of the nonstationary GEV models Indeed, multiple simulations of Markov Chain Monte Carlo (MCMC) techniques based on Bayesian approaches have been employed for extreme surge components (i.e. $\sim$ 1.5-yr, $\sim$ 2-4-yr, $\sim$ 5-8-yr and $\sim$ 12-16-yr provided by the multiresolution wavelet decomposition) to identify the best covariates of climate oscillation to be used for parametrizing the nonstationary GEV models. The most of simulations has mainly supported the results outlined in the previous section: the $\sim$ 1.5-yr of SLP, $\sim$ 2-4-yr of ZW, $\sim$ 5-8-yr of NAO and $\sim$ 12-16-yr of AMO oscillations are considered as the best covariates for modelling respectively the $\sim$ 1.5-yr, $\sim$ 2-4-yr, $\sim$ 5-8-yr and $\sim$ 12-16-yr of monthly extreme surges

3.2. Variable selection.

Table 2 is used to select the most appropriate climate variables to be integrated in the EVA. Though informative and useful to support discussion, my concern is that this selection is mainly based on a correlation analysis (Figure 7 and following ones), and I wonder why the authors did not perform a variable selection for the GEV model directly; for instance using AIC or selection criteria. See a discussion by Wong (2018)

Thank you for this comment.

Indeed and as suggested in the part 3.1 of the present document, the use of the climate index as a covariable in the GEV model has been well justified by (1) the wavelet

coherence (Table 2) and (2) a Bayesian approach has been used also for assessing extremes in a changing climate oscillation (NAO, SLP, ZW and AMO) at each timescale by making inferences from the Likelihood function (validation of the first hypothesis). Once the climate covariate has been selected, the AIC criteria have been used for the implementation of the best use of climate index onto the GEV parameters. This part needs to be more explained in the new version. More clarifications related to this point have been added (lines 541-560 in the new version of the manuscript).

The connections between the climate oscillations and the monthly maxima at the different timescales (Figure 9)., presented previously (section 5.1), have been explored as a first hypothesis for the implementation of the nonstationary GEV models Indeed, multiple simulations of Markov Chain Monte Carlo (MCMC) techniques based on Bayesian approaches have been employed for extreme surge components (i.e. $\sim$ 1.5-yr, $\sim$ 2-4-yr, $\sim$ 5-8-yr and $\sim$ 12-16-yr provided by the multiresolution wavelet decomposition) to identify the best covariates of climate oscillation to be used for parametrizing the nonstationary GEV models. The most of simulations has mainly supported the results outlined in the previous section: the $\sim$ 1.5-yr of SLP, $\sim$ 2-4-yr of ZW, $\sim$ 5-8-yr of NAO and $\sim$ 12-16-yr of AMO oscillations are considered as the best covariates for modelling respectively the $\sim$ 1.5-yr, $\sim$ 2-4-yr, $\sim$ 5-8-yr and $\sim$ 12-16-yr of monthly extreme surges. Once the climate covariate has been selected for each timescale, three nonstationary models have been used by introducing the climate information as a covariate into: (1) the location parameter (GEV1); (2) both location and scale parameters (GEV2); (3) all location, scale and shape parameters (GEV3). The structure of the most appropriate nonstationary GEV distribution has been selected by choosing the most adequate parametrization that minimizes the Akaike information criterion (Akaike, 1974). The goodness of fit for each model has been checked through the visual inspection of the quantile-quantile (Q-Q) plots (Figure 10); these plots compare the empirical quantiles against the quantiles of the fitted model. Any substantial departure from the diagonal indicates inadequacy of the GEV model. 3.3 Model selection. Furthermore, the results for Brest in Table 3 may raise some questions:

- For scale_12-16 years, GEV0 does not seem to be the model that leads to the minimum AIC value (-1258 to be compared to -1980 for GEV1);

- For scale _ 2-4-yr, the AIC values

fro GEV1-3 are very close, which make very hard to identify with high confidence the most appropriate model. The authors should comment on that. See also Burnham and Anderson (2004) for further details. Reference: Wong, T. E. (2018). An integration and assessment of multiple covariates of nonstationary storm surge statistical behavior by Bayesian model averaging. Advances in Statistical Climatology, Meteorology and Oceanography, 4(1/2), 53-63.

Burnham, K. P. and Anderson, D. R.: Multimodel inference: understanding AIC and BIC in model selection, Sociolog. Meth. Res., 60, 261–304, 2004.

It's a very interesting comment which needs more clarifications from the authors. More discussion related to this part has been added basing on the references provided. Also, the different results presented here still preliminary and represent a first step for investigating the nonstationary behavior of the different frequencies. In the light of the present results, the nonstationary behavior is mainly controlled by the high frequencies.

More discussion related to the stationarity of the low frequencies ($\sim$ 12-16 years) ; lines 610 -617.

[revised manuscript text omitted]

This part has been added in the manuscript (lines 415 -420). For each timescale, a bootstrap approach has been applied to assess the statistical significance of the correlation between the spectral component of the extreme surges and the climate oscillation. By resampling the timeseries 10.000 times, 95% confidence intervals have been considered to extract the best climate information fitting the extreme surges (Villarini et al., 2009). 5. Typo. Line 70: "investigates" should be "investigate" Line 467: "covariable" should be covariate All typos have been checked and corrected.

Please also note the supplement to this comment:
https://nhess.copernicus.org/preprints/nhess-2020-101/nhess-2020-101-AC2-supplement.pdf
* * *
[Figure]

[Figure]

**Fig. 1.** Figure 1 Geographical location of the study area and the different tide gauges along the English Channel coasts: Brest, Cherbourg, Dunkirk (NW France); Dover and Weymouth (SW UK).

[Figure]

**Fig. 2.** Figure 2. CWT of monthly maxima of surges in Brest, Cherbourg, Dunkirk, Dover and Weymouth.

[Figure]

**Fig. 3.** Figure 3. Multiscale variability of the monthly mean and maximum surges in Brest. (a)
CWT of monthly mean surges; (b) Interannual variability of monthly and extreme surges

[Figure]

**Fig. 4.** Figure 4 Wavelet details (components) resulting from the multiresolution analysis of surges at the interannual ($\sim$ 1.5-yr , $\sim$2-4-yr and $\sim$5-8-yr) and interdecadal ($\sim$12-16-yr) time scales for all sites (

[Figure]

**Fig. 5.** Figure 5. Coherence-wavelet diagrams between monthly extrema of surges and Sea Level Pressure (SLP).

[Figure]

**Fig. 6.** Figure 6. Coherence-wavelet diagrams between monthly extrema of surges and Zonal Wind (ZW).

[Figure]

**Fig. 7.** Figure 7. Coherence-wavelet diagrams between monthly extrema of surges and North Atlantic Oscillation (NAO).

[Figure]

**Fig. 8.** Figure 8. Coherence-wavelet diagrams between monthly extrema of surges and Atlantic Multidecadal Oscillation (AMO).

[Figure]

**Fig. 9.** Figure 9 Wavelet details of monthly extreme surges (black lines), at the interannual (∼ 1.5-yr , ∼2-4-yr and ∼5-8-yr) and interdecadal (∼12-16-yr) time scales for all sites (Brest, Cherbourg, Dunkirk

[Figure]

**Fig. 10.** Figure 10. a. The quantile plot between observed and modelled extreme surges by the use of the best GEV models, at different time scales, case of Brest. b. The Return level of extreme surges estimated

---

## Author Response (AR1)

**Reviewer 1 :**

**General Suggestions**

\# Review on the paper "A nonstationary analysis for investigating the multiscale variability of extreme surges: case of the English Channel coasts"

\#\# General comments

In the research manuscript, the authors present an analysis of time series of storm surges in five stations along the coast of the English Channel.

The study is two-fold: a first part is related to the analysis of the monthly extreme storm-surges signals, by using a multi-scale wavelet analysis in order to describe the

**Answer to Reviewer 1 :**

Thank you for the different suggestions and comments useful for the improvement of the manuscript.

The text of **the manuscript has been checked by the authors and has been simplified to make easier the writing and the understanding of the different sections, in particular the** discussion and the methodology.

Also, some information has been moved from the methodological part to the results. Two illustrations related to the original hydrodynamic data (Figure 4 in the new version) and the morphological defects (Figure 2 in the new version).

The answers will be addressed for each specific comment.

**### 2. Data**

It no very clear in the paper if storm surge or the sea level height data is used. Only the later is measured by the tide gauge, and thus a pre-processing step is required to filter out the tides and the sea-level rise. The description of this pre-processing seems to be missing in the paper. Another question is about the availability of the large-scale atmospheric circulations indices (NAO, AMO...) during the whole period covered by the tide gauges. In particular, the Brest station has measurements from 1850, so one could wonder if the indices are available from the date, and what could be the quality of such values. I think that the paper could benefit from some discussion on this point.

**Answer**
*Thank you for this suggestion.*

*According to the determination of surges, a new part explaining this extraction has been*
*added in the new version: part 3.1;*

*According to the availability of the large-scale atmospheric circulations indices (NAO,*
*AMO…), they have been extracted from the NCEP-NCAR Reanalysis fields with the same*
*period of the surges.*

*Timeseries of surges*

*Weymouth: 1991-2018*

*Brest: 1846-2018*

*Dunkirk:1964-2018*

*Dover: 1958-2018*

*Cherbourg: 1964-2018*

*Timeseries of the correlations Climate index - maximum surges*

*The correlation has been carried depending on the length of the*

*AMO/BREST: 1880 – 2017*

*NAO /Brest : 1865-2017*

*SLP /Brest : 1948-2017*

*ZW /Brest : 1865-2017*

*AMO – NAO – SLP - ZW/ Cherbourg - Dunkirk : 1964-2017*

*AMO – NAO – SLP - ZW/Douvres : 1958-2017*

*AMO – NAO – SLP- ZW/Weymouth : 1991-2017*

*The methodological approach has benefit from this information.*

### 3. Extreme value models
The authors use the classical extreme value distribution (GEV) to model the
monthly maxima of storm surges, making the distribution non-stationary by
incorporating climate indices as covariates.

Although the presentation of the model is rather clear, the data on with the model is applied is not as clear to me: is it on the initial time series of storm surges or on the spectral components? On L200, it seems that the model is applied to each spectral component, but the justification of using a GEV distribution is then questionable since the component by itself is not extreme nor a maxima, and thus an extreme value distribution is not justified. Marginal distributions of the variable on which the GEV is fitted could give some insight on the adequacy of a GEV, in addition to the QQ plots of Figure 10.

I have another remark about model selection: the authors do not show the fitted parameters values nor the associated confidence intervals, but only the AIC values in Table 3. Such values would be necessary to address the fit and to discuss whether or not the influences of the indices are significant. The authors are only selecting the parameter that cannot be considered are stationary, but not the index that is relevant to explain the non-stationary.

**Answer**

*The authors try to add more clarifications related to the application of the GEV model.*

*Indeed, the maxima of surges has been decomposed to different frequencies to which the model has been applied. This approach has been applied by Turki et al., 2019 ; 2020. The low and the high frequencies of the maxima highlight the different fluctuations of the signal and their multiscale variability. The prediction of each fluctuation has been investigated by the use of nonstationary GEV model with the incorporation of climate indices.*

**### 4. Multi-timescale variability of extreme surges**

The authors describe the results of the continuous wavelet transform (CWT) on the monthly maxima of storm surges to assess the non-stationary behavior;

**Answer**

*The authors have used the CWT to identify the spectrum of the extreme signal and its distribution in space (between station) and time (during the period of study); (Figure 2).*

*The authors have shown also (Fig 3) that the low frequencies ( lower than 10 year) are clearly observed from the CWT of the extreme surges, and less identified from the mean surges.*

*In this part, a quantification of the spectral components has been presented (Figure 4).*

**5. Large-scale climate North-Atlantic oscillations and their link to extreme surges in the English Channel**

This section is two-fold : first, exhibit the link between the indices and the monthly maxima of storm surges and fit the GEV distribution to the components of the storm surge.

The authors look at the wavelet coherence to address this question and conclude that:

Each timescale exhibits mainly strong links with its associated climate index (L313)

Such a conclusion seems rather obvious to me because the indices are constructed that way and is not sufficient to my point of view to do variable selection in the GEV model of the following section before fitting the model.

Although the lengthy discussion about the visual inspection is interesting and may worth it, a proper statistical method to select the variable should be preferred.

Once the model is fitted, the paper falls short : since there is no variable selection and no use of the fitted model, what is the fit used for? We only can see with some difficulties the return levels of each component for the Brest station, but with little extrapolation.

As is, the relevance of using a GEV model is questionable.

**Answer**

*Thank you for these suggestions.*

*Indeed, according to the previous works, the effects of the climate patterns is important and should be considered for predicting the variability of extreme surges.*

*In the present research, the novel approach that an excellent prediction of the total signal requires a good multi-timescale prediction, i.e a prediction of each spectral component (provided by the MODWT analysis) which is described by an appropriate climate index ( the most appropriate one has been selected basing on the wavelet coherence and the monte carlo iterations for each component).*

*Indeed, the non-linear interaction between the physical mechanisms of climate patterns is very complex and could not considered at the same time to predict the extreme surges. To investigate this interaction our hypothesis consists on developing a statistical model able to predict the spectral component with the incorporation of the most adequate climate information.*

*The development of a full model useful for estimating the extreme surges needs the integration of the GEV models associated to the different timescales (~ 2-4 years; ~ 5-8 years; ~ 12-16 years) by the means of mathematical methods; which is the objective of the further works. The present work bring a novel hypothesis to resolve the complex effects of climate patterns on the local variability of surges. This step is very important to introduce a new model considering the different timescales of extreme surges.*

**Answer**

*All figures have been improved in terms of quality.*

**## Technical remarks**

- Spacing is not uniform in the text, please consider proof-reading;
- Parameters values are missing in Table 3;
- a ½u on L805; - Brest is missing in Figure 1; On L150:
*The observations which correspond to the hydrographic zero level are referenced to zero tide gauge (Figure 1)
* but seems missing;
- Figures 2-3-5-6-7-8 : the time-scales can hardly be seen, please choose another color or another representation or another colormap.
- Figure 2 color bar is not coherent with the other Figures;
- Figures 4 and 9: the x-axis is not clear, at least 4-digit years should be provided.
- Figure 10 a : The y-axis should be "Modeled values";
- Figure 10 b: the figure can hardly be read, please provide a larger version or a better quality (e.g. SVG)

**Reviewer 2 :**

**General Suggestions**

The manuscript by Turki and co-authors addresses an important issue for the modeling of exceedance probability of extreme surges namely accounting for the dependence with climate patterns. The authors present an approach relying on wavelet analysis to investigate the correlation between the extreme surges and four climate oscillations (North Atlantic Oscillation, and Atlantic Multidecadal Oscillation, and the ones related to Sea-Level Pressure and Zonal Wind) at multiple time scales, _1.5-years, _2-4-years, and _5-8-years and 12-16-years. On this basis, they perform nonstationary extreme value analysis using the English Channel coasts as application cases and show the added-value for accounting for these multiscale processes when deriving the return periods.

Main comment

The manuscript is well organized and the presentation is clear. Yet, several
aspects should be clarified and further elaborated before publication (state
of the art, details of the implementation, discussion regarding the
assumptions). Therefore, I recommend additional corrections by
incorporating, if possible, the following recommendations.

**Answer to Reviewer 2 :**

**Specific comments**

**1. State of the art.**

Some key references about the link between extreme surges and climate
variables should be added to the bibliography, namely:

** Marcos, M., Calafat, F.M., Berihuete, Á., Dangendorf, S., 2015. Long term
variations in global sea level extremes. J. Geophys. Res. 120(12), 8115-8134.

** Marcos, M.; Woodworth, P.L., 2017. Spatiotemporal
changes in extreme sea levels along the coasts of the North Atlantic and
the Gulf of Mexico. J. Geophys. Res., 122(9), 7031-7048.

** Méndez, F. J., Menéndez,M., Luceño, A., Losada, I. J., 2007. Analyzing
monthly extreme sea levels with a time-dependent GEV model. Journal of
Atmospheric and Oceanic Technology 24(5), 894-911.

** Wahl, T., Chambers, D.P., 2015. Evidence for multidecadal variability in US
extreme sea level records. J. Geophys. Res. 120, 1527–1544.

** Wahl, T.; Chambers,D.P., 2016. Climate controls multidecadal variability in
US extreme sea level records. J. Geophys. Res. 121(2), 1274-1290.

My second concern relates to the differences of the present work with the
recently published one, namely Turki et al. (2020). As far as I understood, the
time scale 12- 16-years and the British part of the Channel coasts were not
tackled in this published work, but it would be useful to situate in more details
the present study with respect to Turki et al. (2020), for instance in the
introduction.

*Thank you for the comments.*

*The proposed references have been revised and added to the state of the art.*

*The new funding proposed in this research, compared to the last work of Turki et al., 2020,*
*has been better explained in the state of the art as suggested in line …………………*

**2. Details on the implementation.**

The authors focus on extreme surges. To do so, the raw data of tide gauges
should be pre-processed by accounting for the tide. Could the authors
provide more details on how they proceeded? What type of tide data did
they used?

Similarly, the authors used climate indices provided by the NCEP-NCAR
Reanalysis.
Could the authors provide the web link where they downloaded the data for
the climate indices? Besides, the authors mentioned climate oscillations
using SLP and Zonal Wind. Are they directly available from NCEP-NCAR
Reanalysis or are they derived from a pre-processing using EOF analysis for
instance?
3. Model selection in the non-stationnary Extreme Value Analysis (EVA).
Integrating the climate drivers as covariates in EVA is a good idea, but the
selection of the 'most appropriate' model deserves more discussion.

*Regarding the extraction of extreme surges, more details related to the classical model used*
*for calculating tides are provided in the new version.*

*Also, the different climate indices have been better explained.*

*The selection of the most appropriate GEV model has been achieved for each frequency*
*component. The use of the climate information has been differently explained for the*
*different spectral component. For each frequency, the different climate indices have been*
*tested to analyze the nonstationary behavior of the component. Then, the climate index*
*associated with the best model fitting data ( i.e. showing the lowest AIC) has been selected*
*and presented in this research. Similar approach has been used in Turki et al., 2020.*

*More explanations related to this part has been added in the new version.*

**3.1. Adequacy of GEV.**

It is not clear to me whether extreme value distributions are applied to each
spectral component. If so, I wonder whether these variables are 'extreme',
and whether GEV distribution is appropriate. Could the authors comment on
that?

*The monthly extreme surges have been calculated from hourly residual sea level. This signal has been decomposed by the MODWT to study separately the different components. Our hypothesis in the present research is the following:*

*The variability of the local extreme surges should be explained by the global climate patterns described by a series of physical mechanisms associated to the climate indices.*

*We have used the hypothesis that each spectral should be explained by a climate mechanism. Such hypothesis has been justified and validated by (1) the coherence diagrams (see also Table 2) where we have demonstrated that the effect of each climate index on the variability of extreme surges varies as a function of the spectral component and (2) the monte carlo analysis applied to each spectral component to select the most appropriate climate index. This analysis has shown a strong coherence with the first validation (1).*

**3.2. Variable selection.**

Table 2 is used to select the most appropriate climate variables to be integrated in the EVA. Though informative and useful to support discussion, my concern is that this selection is mainly based on a correlation analysis (Figure 7 and following ones), and I wonder why the authors did not perform a variable selection for the GEV model directly; for instance using AIC or selection criteria. See a discussion by Wong (2018).

**Answer**

*Thank you for this comment.*

*Here, a Bayesian estimation has been used to make inferences from the Likelihood function. The reason behind the choice of this approach is overcoming the limitation of short time-series with small size, the case of Weymouth station where date cover the period from 1991 and 2018.*

*For each spectral component, a sample of 100.000 length has been modelled by GEV using a selected climate index. The upper and lower quantiles of the posterior probability distribution for the parameters of the MCMC sample are taken. The goodness of fit has been taken as a function of the values of the upper and the lower quantiles; best results have been considered when these values are higher than 92.5% and lower than 5.2%, respectively.*

*A new section has been added in the methodological Approach*

**4 Determination of the most appropriate climate oscillation** 271 *connected to each timescale extreme surges for GEV models*

*Then and once selected, the most appropriate climate index has been incorporated into the GEV covariates (into the different parameters). By the use of AIC criteria, we have the best*

*results of fitting for the incorporation of this index into one (location only), two (location and*
*scale) or three (location, scale or shape) parameters of GEV.*

*Also, More clarifications related to this point have been added in the results (lines 573-581 of*
*the new version pfd)*

**3.3 Model selection.**

Furthermore, the results for Brest in Table 3 may raise some questions:

- For scale_12-16 years, GEV0 does not seem to be the model that leads to the minimum AIC value (-1258 to be compared to -1980 for GEV1);

 - For scale _ 2-4-yr, the AIC values fro GEV1-3 are very close, which make very hard to identify with high confidence the most appropriate model. The authors should comment on that.
See also Burnham and Anderson (2004) for further details.

The authors analyze the significance of the correlation through a visual inspection of the results provided by wavelet spectral analysis. In lines 339-341, the authors mentioned that they are using a Monte-Carlo-based approach to identify the most statistically significant correlation: could the authors provide more details on the implementation.
Is it a bootstrap-based approach? How do they analyse the changes of the correlation at the Monte-Carlo iterations? Could the authors provide
additional results about this significance assessment?
*__Answer__*

*As suggested in the previous document, a bootstrap approach has been applied to assess the*
*statistical significance of the correlation between the spectral component of the extreme*
*surges and the climate oscillation at each timescale. By resampling the timeseries 10.000*
*times, 95% confidence intervals have been considered to extract the best climate information*
*fitting the extreme surges (Villarini et al., 2009).*

*Here, the confidence intervals (CI) have been calculated by the bootstrap technique by*
*simulating the monthly maxima of surges (spectral component) from the climate index*
*(spectral component) at each timescale (new samples with a size of 1000).*

*When the original surges have been fitted to the simulated ones, 95% confidence intervals for*
*the maximum likelihood estimates have been calculated.*

*A table providing the 95% CI for each spectral component and each station is added (Table 3).*

*Table 3 Analysis of the statistical significance of the correlation between the spectral*
*component of the extreme surges and the climate oscillation at each timescale for the*
*different stations. The 95% Confidence Intervals from Bootstrap technique in Square*
*Brackets. The most significant correlations are illustrated by the grey columns.*

| ~ 1.5-yr | SLP | ZW | NAO | AMO |
|---|---|---|---|---|
| **Brest** | [0.152, 0.174] | [0.145, 0.182] | [0.141, 0.178] | [0.138, 0.189] |
| **Cherbourg** | [0.161, 0.170] | [0.142, 0.179] | [0.142, 0.179] | [0.135, 0.180] |
| **Dunkirk** | [0.160, 0.168] | [0.150, 0.185] | [0.150, 0.185] | [0.135, 0.183] |
| **Dover** | [0.158, 0.165] | [0.161, 0.180] | [0.161, 0.180] | [0.133, 0.180] |
| **Weymouth** | [0.421, 0.429] | [0.411, 0.450] | [0.381, 0.299] | [0.375, 0.281] |

**~ 2-4-yr**

| | SLP | ZW | NAO | AMO |
|---|---|---|---|---|
| **Brest** | [0.145, 0.164] | [0.149, 0.158] | [0.141, 0.179] | [0.138, 0.183] |
| **Cherbourg** | [0.160, 0.175] | [0.188, 0.196] | [0.161, 0.179] | [0.158, 0.182] |
| **Dunkirk** | [0.145, 0.158] | [0.180, 0.185] | [0.145, 0.164] | [0.140, 0.169] |
| **Dover** | [0.148, 0.163] | [0.192, 0.198] | [0.145, 0.168] | [0.143, 0.175] |
| **Weymouth** | [0.412, 0.420] | [0.420, 0.430] | [0.410, 0.425] | [0.410, 0.428] |

**~ 5-8-yr**

| | SLP | ZW | NAO | AMO |
|---|---|---|---|---|
| **Brest** | [0.075, 0.090] | [0.073, 0.092] | [0.085, 0.089] | [0.070, 0.096] |
| **Cherbourg** | [0.190, 0.198] | [0.185, 0.198] | [0.191, 0.196] | [0.181, 0.198] |
| **Dunkirk** | [0.180, 0.188] | [0.177, 0.185] | [0.183, 0.187] | [0.175, 0.187] |
| **Dover** | [0.180, 0.195] | [0.180, 0.198] | [0.180, 0.184] | [0.176, 0.199] |
| **Weymouth** | [0.219, 0.222] | [0.218, 0.225] | [0.221, 0.226] | [0.216, 0.226] |

**~ 12-16-yr**

| | SLP | ZW | NAO | AMO |
|---|---|---|---|---|
| **Brest** | [0.033, 0.046] | [0.034, 0.045] | [0.035, 0.045] | [0.038, 0.041] |
| **Cherbourg** | [0.089, 0,099] | [0.090, 0,099] | [0.090, 0,097] | [0.091, 0,095] |
| **Dunkirk** | [0.087, 0.099] | [0.089, 0.098] | [0.090, 0.097] | [0.093, 0.096] |
| **Dover** | [0.078, 0.089] | [0.080, 0.088] | [0.080, 0.086] | [0.082, 0.085] |
| **Weymouth** | [0.250, 0.260] | [0.250, 0.259] | [0.250, 0.257] | |

**Model for each Spectral component?**

The objective of this decomposition and the use of GEV par component.
The full model for estimating the extreme surges can be the sum of the different components?

**Answer**

*In any part of the manuscript, the authors have proposed the sum of the different GEV models to obtain the full model able to estimate the total signal of monthly extreme surges.*

*Indeed, this question remains unresolved until now. Once the model identified, how can we use them to define a stochastic tool for estimating the extreme surges?*

*However, this finding confirms our first hypotheses related to the connection between the most appropriate climate index and the extreme surges at each time scale. It provides also a guidance on incorporating nonstationary processes of large-scale oscillations to different spectral components informed by the wavelet techniques, the Bayesian approaches and the GEV model probabilities.*

*A new paragraph explaining this point has been added as a discussion part.*

This study has expanded the previous works of Turki et al. (2019; 2020) upon a new approach combining spectral and probabilistic methods to integrate multiple streams of information related to climate teleconnections. Indeed, each timescale has been simulated separately with the nonstationary GEV models and expressed as a function of the most suitable climate index improving its fitting. The estimation of the total signal of surges should be determined by combining the developed nonstationary GEV models used for the different timescales.

These results should support the hypothesis introduced at the beginning of the present work suggesting that: (i) the extreme surges should depend on different timescales; (ii) each timescale should be related to a specific large-scale oscillation.

The finding is in agreement with the previous works of Lee et al. (2017) and Wang et al. (2018) highlighting the importance of a careful consideration when complex physical mechanisms of different climate indices are included into model structures for estimating extreme surges.

Indeed, this work provides a guidance on incorporating nonstationary processes of large-scale oscillations to different spectral components informed by the wavelet techniques, the Bayesian approaches and the GEV model probabilities.

The primary contribution of the present research is to present a new approach for: (1) investigating the multi-timescale variability of the nonstationary extreme surges; (2) identifying their multi-connection with climate oscillations according to the timescale and (3) resolve in part the problems of uncertainty of most appropriate climate to use as covariate for GEV models at each timescale. However, additional models (e.g. significance tests and sensitivity analyses and modelling uncertainties) and application sites (e.g. Mediterranean and pacific ones controlled by other climate oscillations) are required to expand the developed approach.

Also, generating a final robust stochastic model useful for projecting storm surge return levels and assessing the flood risk management requires further efforts to build on the potentially advantageous approach presented here by integrating the GEV models associated with the different timescales through the use of mathematical methods.

**Figures for comparing the return levels.**

**Answer**

*Here, the stochastic modelling provides us some insights into the nonstationary behavior of extremes in relation with the climate oscillations.*

*The comparison between the simulated surges, using the nonstationary GEV models, and the original data has been illustrated in Fig.10 for Brest.*

*The fifty-year return level plot for the different spectral components is presented in Fig.10 c.*

[Figure]

*Figure 10.c Fifty-year return level of monthly values using the original data (grey circles) and the best nonstationary GEV model for Brest (solid black line) at each timescale (spectral component). The lower and the upper limits of the 95% confidence interval calculated using the delta method (dashed black line). The associated confidence area is plotted with grey shaded area.*

**5. Typo.**

Line 70: "investigates" should be "investigate" Line 467: "covariable" should be covariate

*All typos have been checked and corrected.*

[revised manuscript text omitted]

---

## Author Response (AR2)

**Comments for Reviewer**

*Dear Reviewer*

*Thanks again for your constrictive comments to make easier the reading of the work and clearer the different steps implemented.*

*Your comments suggested in the first and the second revision were very useful to put this work clearer and easy to understand. Thank you for the time spent and the interest.*

*As suggested in previously, the work presents preliminary results which are very promising. This approach will be strengthened and more investigated in the framework of a current PhD project related to extreme dynamics and also other european collaborations/projects*

**Comment 3.**

*Indeed, the methodology is hard to follow since we have mentioned all techniques used which is very confusing.*

*The idea suggested in your third comment is really excellent to make the methodological approach easier to understand.: flowchart summarizing the steps.*

*A figure 2 has been prepared and added in the manuscript to summarize the different steps implemented and the methods used (answer to comment 3).*

*Also, a detailed description related to the methodology has been added in the section 3.4*

**Lines 272- 312 in the new version of the manuscript (the yellow part).**

**3. 4 Determination of the most appropriate climate oscillation connected to each timescale extreme surges for GEV models**

As suggested previously, the main hypothesis presented in this research is that effects of the physical mechanisms on the extreme surges vary according to the timescale and each scale should be related to a given climate oscillation.

This hypothesis has been investigated by two approaches:

(1) a spectral approach based on the use of wavelet techniques (wavelet multiresolution and wavelet coherence as detailed in section 3.2) for optimizing the physical relationship between the climate index and the extreme surges at each timescale.

The Bootstrap is a resampling technique used to estimate the sampling distribution of an estimator of sample statistics by drawing randomly with replacement from a set of data points.

Here, a bootstrap approach has been applied to assess the statistical significance of the correlation between the spectral component of the extreme surges and the climate oscillation at each timescale. By resampling the timeseries 10.000 times, the extreme surges have been simulated and compared to the original records; the 95% confidence intervals have been considered to extract the best climate information fitting the extreme surges (Villarini et al., 2009).

(2) a Bayesian estimation has been used to make inferences from the Likelihood function. The reason behind the choice of this approach is overcoming the limitation of short time-series with small size, the case of Weymouth station where the measurements covers the period from 1991 and 2018.  A technique of Markov Chain Monte Carlo (MCMC), implemented in the evbayes package within R software, has been used basing on multiple simulations (the number of simulations is varying as a function of the length of the timeseries).

For each spectral component, a sample of 100.000 simulations has been modelled by GEV using a given climate index. The upper and lower quantiles of the posterior probability distribution for the parameters of the MCMC sample are taken. The goodness of fit has been taken as a function of the values of the upper and the lower quantiles; best results have been considered when these values are higher than 92.5% and lower than 5.2%, respectively.

Both approaches have been used to select the most appropriate atmospheric physical mechanism to each timescale of extreme surges. Then and at each timescale (i.e. spectral component), the selected mechanism (i.e. the climate oscillation in this case) has been used as covariate for modelling the extreme surge by nonstationary the implemented GEV models.  The

best use of the covariate into the different GEV parameters (location, scale and shape) have been investigated by means of AIC criterion.

Once the best GEV models defined for each time scale, a series of simulations have been carried out to compare modelled and observed surges.

This Bayesian inference has been also used to calculate: (1) the return levels of the nonstationary simulated surges which were compared to those of the observed ones; (2) the confidence interval (CI) assessing the goodness of this comparison.

Figure 2 summarizes the methodological approach proposed in the present research and the different steps implemented. The statistical methods used to resolve each step are also synthetized in this figure.

*Figure 2. A summary of the methodological approach implemented in this research.*

[Figure]

HYPOTHESES:
1. Effects of the atmospheric physical mechanisms on the extreme surges vary according to a given timescale.
2. Each timescale should be related to a given climate oscillation.

*Figure 2. A summary of the methodological approach implemented in this research.*

**Comment 2.**

*The values extracted from the bootstrap technique have been double checked. We are very sorry for the inversion of the upper and lower interval for Weymouth (error when we have imported the values to the table).  The corrected results are shown in Table 3.*

[revised manuscript text omitted]

**Comment 3.**

*Thank you for this comment. Effectively many implementations for applying the same approach is very confusing.*

*This part has been clarified in the new version of the manuscript.*

*We focus only on the use of the Bayesian approach for defining the best climate covariate of each time scale and also for calculating the return levels.*

**Lines 306-308 of the new version:**

This Bayesian inference has been also used to calculate: (1) the return levels of the nonstationary simulated surges which were compared to those of the observed ones; (2) the confidence interval (CI) assessing the goodness of this comparison.

This point has been also illustrated in the Figure 2.